# Incorporating Surrogate Gradient Norm to Improve Offline Optimization Techniques

**Manh Cuong Dao, Phi Le Nguyen**
Hanoi University of Science and Technology
Cuong.DM242249M@sis.hust.edu.vn,lenp@soict.hust.edu.vn

**Thao Nguyen Truong**
National Institute of Advanced Industrial Science and Technology
nguyen.truong@aist.go.jp

**Trong Nghia Hoang**[*]
Washington State University
trongnghia.hoang@wsu.edu

## Abstract

Offline optimization has recently emerged as an increasingly popular approach to mitigate the prohibitively expensive cost of online experimentation. The key idea is to learn a surrogate of the black-box function that underlines the target experiment using a static (offline) dataset of its previous input-output queries. Such an approach is, however, fraught with an out-of-distribution issue where the learned surrogate becomes inaccurate outside the offline data regimes. To mitigate this, existing offline optimizers have proposed numerous conditioning techniques to prevent the learned surrogate from being too erratic. Nonetheless, such conditioning strategies are often specific to particular surrogate or search models, which might not generalize to a different model choice. This motivates us to develop a model-agnostic approach instead, which incorporates a notion of model sharpness into the training loss of the surrogate as a regularizer. Our approach is supported by a new theoretical analysis demonstrating that reducing surrogate sharpness on the offline dataset provably reduces its generalized sharpness on unseen data. Our analysis extends existing theories from bounding generalized prediction loss (on unseen data) with loss sharpness to bounding the worst-case generalized surrogate sharpness with its empirical estimate on training data, providing a new perspective on sharpness regularization. Our extensive experimentation on a diverse range of optimization tasks also shows that reducing surrogate sharpness often leads to significant improvement, marking (up to) a noticeable $9.6\%$ performance boost. Our code is publicly available at https://github.com/cuong-dm/IGNITE.

## 1 Introduction

A central task in numerous scientific disciplines is to optimize for some material configuration that maximizes a certain utility metric. Previously, this would incur an expensive and repetitive experiment process that requires a huge amount of human-labor. To bypass such inefficiencies, a data-driven approach [1, 2, 3, 4, 5, 6, 7, 8, 9, 10] has recently been adopted. Instead of laboring on a new set of expensive on-demand experimentation for each new optimization task, we can leverage past experimentation data to build a parametric model that predicts the outcome of the experiments themselves. Its parameters are tuned to fit past experimental data and then fixed while optimizing for the best input.

---

[*]Corresponding authors: Manh Cuong Dao, Trong Nghia Hoang.

38th Conference on Neural Information Processing Systems (NeurIPS 2024).

However, in most applications, offline data is rarely representative of the entire input space. As a result, the surrogate model's prediction is often not accurate outside the offline data regime, potentially overestimating the outputs at sub-optimal input candidates. To mitigate this, existing offline optimizers have introduced numerous regularizing strategies for either the surrogate or the search models. For example, [4, 11, 12, 13] and [3] regularize their surrogate models so that their predictions at inputs outside the offline data regime are pushed down or pushed towards a constant value. Alternatively, [1] and [2] restrict their search models to regions having certain domain-specific properties under which sampled inputs probably have high performance.

These strategies therefore depend on the specifics of either the surrogate- or the search procedures to characterize the out-of-distribution regions or the desirable domain-specific properties. Consequently, their regularization might not extend to out-of-distribution regions which are not sufficiently specified. For example, COMs [13] uses an ad-hoc specification that characterizes the out-of-distribution region in terms of inputs that are reached during the first few iterations of gradient ascent on an un-regularized surrogate. This only characterizes a local sub-region of a broader out-of-distribution data regime.

To mitigate such limitations in the regularizing behaviors of prior work, we instead aim to investigate a more generic approach that is independent of the specifics of the search or surrogate procedures. For instance, instead of regularizing the behavior of the model on certain input regions, we could impose constraints on the geometries of its loss landscape such as requiring it to be in a parameter regime that uniformly produces low loss values [14]. Such regularizing strategies can, therefore, be incorporated into existing offline optimizers as an additional regularizer to improve their performance. To substantiate this idea, we have made the following technical contributions:

**1.** We develop a model-agnostic regularizer (for surrogate training) based on a notion of model sharpness[2]. This is characterized in terms of the surrogate's maximum output change under low-energy parameter perturbation (i.e., norm-bound perturbation) (Section 3.1). Intuitively, suppose a surrogate's prediction does not change substantially within a perturbation neighborhood (i.e., via adding perturbation to its parameters) that contains the oracle, its predictions are likely to be close to those of the oracle (see Fig. 1). As such, minimizing this sharpness measure can help suppress the erratic behavior of the surrogate model at out-of-distribution input.

**2.** We adopt a practical approximation that interestingly reduces the above surrogate sharpness measurement to a function of the surrogate's gradient norm. The surrogate training can then be augmented into a constrained optimization task whose constraint imposes a user-specified threshold on the surrogate's gradient norm. We can then solve it to acquire the optimally regularized surrogate using existing constrained optimization solvers. The high-level pseudo-code of our proposed algorithms which incorporate surrogate gradient norms to improve existing offline optimization techniques (**IGNITE**) is detailed in Algorithm 1 (Section 3.2).

**3.** We develop a detailed theoretical analysis to show that reducing surrogate sharpness on the offline dataset provably reduces its generalized sharpness on unseen data. Our analysis extends existing theories [14] from bounding generalized prediction loss (on unseen data) with loss sharpness to bounding the worst-case generalized surrogate sharpness with its empirical estimate on training data, providing a new perspective on sharpness regularization (Section 4).

**4.** We demonstrate empirically that incorporating the proposed model-agnostic regularizer into existing offline optimizers as an additional conditioning component often results in significant improvement over existing offline optimizers in most cases. This sets the first step towards a new, synergistic research direction in offline optimization that can potentially support and complement both existing and future work (Section 5).

For interested readers, a concise review of existing literature is also provided in Section 2.

## 2    Problem Definition and Related Works

In this section, we will concisely review the preliminaries of offline optimization. Section 2.1 provides a mathematical formulation of offline optimization, and Section 2.2 summarizes prior works.

---

[2]Our sharpness notion is different from that of [14] which is applied to the training loss. Instead, our sharpness measure is accessed on the surrogate prediction. Empirically, we also notice that surrogate sharpness often appears to be more effective than loss sharpness in boosting the offline optimization performance (Section 5.3).

## 2.1 Problem Definition

Offline optimization is a computational approach to a variety of material engineering tasks that aim to find a material construction or design that maximizes certain desirable properties. Mathematically, we assume that there is an oracle function $g(\mathbf{x})$ that maps from a material design $\mathbf{x} \in \mathcal{X}$ to an overall utility $z = g(\mathbf{x})$ of its property measurements; and we need to find its maxima:

$$\mathbf{x}_* \quad \triangleq \quad \underset{\mathbf{x} \in \mathcal{X}}{\arg\max} \ g(\mathbf{x}) \ . \tag{1}$$

However, the key challenge here is that $g(\mathbf{x})$ is inaccessible. Instead, we only have access to an offline dataset of observations $\mathcal{D} = \{(\mathbf{x}_i, z_i)\}_{i=1}^n$ where $z_i = g(\mathbf{x}_i)$, which denote the past input-output queries extracted from $g(\mathbf{x})$ in previous experiments. A direct approach to this problem is to learn a surrogate $g(\mathbf{x}; \boldsymbol{\omega}_*)$ of $g(\mathbf{x})$ via fitting its parameter $\boldsymbol{\omega}_*$ to the offline dataset,

$$\boldsymbol{\omega}_* \quad \triangleq \quad \underset{\boldsymbol{\omega}}{\arg\min} \, \mathcal{L}_{\mathcal{D}}(\boldsymbol{\omega}) \quad \triangleq \quad \underset{\boldsymbol{\omega}}{\arg\min} \sum_{i=1}^n \ell\Big(g(\mathbf{x}_i; \boldsymbol{\omega}), \ z_i\Big) \ , \tag{2}$$

where $\boldsymbol{\omega}$ denotes a parameter candidate of the surrogate and $\ell(g(\mathbf{x}; \boldsymbol{\omega}), z)$ denotes the prediction loss of $g(.; \boldsymbol{\omega})$ on $\mathbf{x}$ if its oracle output is $z$. The (oracle) maxima of $g(\mathbf{x})$ is then approximated via,

$$\mathbf{x}_* \quad \triangleq \quad \underset{\mathbf{x} \in \mathcal{X}}{\arg\max} \ g(\mathbf{x}; \boldsymbol{\omega}_*) \ . \tag{3}$$

Suppose the surrogate $g(\mathbf{x}; \boldsymbol{\omega}_*)$'s prediction is sufficiently accurate over the entire input space, solving Eq. (3) is all we need. However, in most cases, $g(\mathbf{x}; \boldsymbol{\omega}_*)$ often predicts erratically outside the offline data regime, which in turn misleads the optimization towards sub-optimal candidates. To mitigate this, numerous surrogate or search regularizers have been proposed, as summarized next.

## 2.2 Related Works

Most existing offline optimization methods have focused on regularizing either the search or the (surrogate) training procedures. The main focus is to either (1) avoid exploring input regions where the surrogate's prediction is not reliable or (2) regularize the prediction behavior of the surrogate at out-of-distribution input regimes. For example, to regularize the surrogate's prediction, [13] uses input candidates found during the first few iterations of gradient updates on un-regularized models to characterize the out-of-distribution regime. The surrogate can then be re-trained with an additional regularizer that penalizes high-value predictions at those sampled input candidates.

Alternatively, [11] maximizes the normalized data likelihood to reduce prediction uncertainty, which also helps suppress erratic prediction at out-of-distribution regime. Existing techniques in model pre-training and adaptation [3] or transfer learning via co-teaching [15] can also be leveraged to enforce criteria of local smoothness that encodes a preference for conservative prediction to avoid overestimating the oracle output. Otherwise, to regularize the search procedure, [1] and [2] focus instead on learning a generative model of input candidate conditioned their oracle performance. Input candidates that likely achieve high performance can then be synthesized via conditioning the generative procedure on high-value oracle output. [1] characterizes such conditioned distribution via an adversarial zero-sum game. [12] learns a direct inverse mapping from the performance output to the input design using conditional generative adversarial network [16].

Despite their reported successes, these approaches are still limited by their ad-hoc characterization of the out-of-distribution regime. As discussed previously in Section 1, the existing characterization of out-of-distribution input is often based on the specifics of either the surrogate or the search procedures, which are not guaranteed to sufficiently characterize the entire out-of-distribution data regime. This motivates us to develop a more generic out-of-distribution characterization (Section 3) that is external to both the search and surrogate models. Such an approach can be readily incorporated into most existing offline optimizers to boost their performance (Section 5).

## 3 Surrogate Regularization with Sharpness Constraint

This section introduces an additional constraint that transforms the original surrogate optimization in Eq. (2) (Section 3.1) into a constrained optimization problem (COP). The constraint imposes

a user-specified upper-bound on the sharpness of the surrogate. Our proposed formulation draws inspiration from a prior work [14] that aims to minimize the sharpness of the loss function to improve generalization. However, unlike the original work in [14], where the sharpness concept is applied to the loss function, we adapt it for the surrogate's prediction. We find it more suitable since offline optimization's search procedure operates on the surrogate landscape instead of the loss landscape. Moreover, while minimizing loss sharpness [14] can ensure low error for single predictions in the OOD regime, errors may accumulate over consecutive predictions in a gradient-based search. Our insight in Section 3.1 (Fig. 1) suggests that such error accumulation can be mitigated by keeping the surrogate sharpness small during training. Furthermore, we show that the sharpness can be practically approximated in terms of the surrogate's gradient norm, which is more tractable. This allows for direct adoption of existing constrained optimization algorithms [17] to effectively solve for the desired optimally regularized surrogate (Section 3.2).

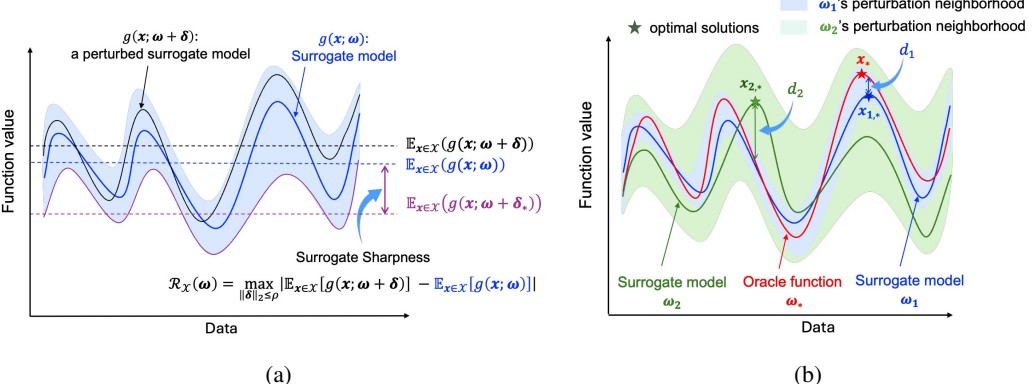

Figure 1: (a) Illustration of surrogate sharpness; (b) Illustration of surrogate sharpness-based offline optimization: Consider two surrogate parameters $\boldsymbol{\omega}_1$ and $\boldsymbol{\omega}_2$ where $\boldsymbol{\omega}_1$ has a smaller sharpness than $\boldsymbol{\omega}_2$. This means the predictions of the models in the perturbation neighborhood of $\boldsymbol{\omega}_1$ will vary less than those of the models in the perturbation neighborhood of $\boldsymbol{\omega}_2$. As such, if both neighborhoods contain the oracle, the prediction error $d_1$ of $\boldsymbol{\omega}_1$ is potentially smaller than the prediction error $d_2$ of $\boldsymbol{\omega}_2$. Consequently, the optimal value of $g(\mathbf{x}; \boldsymbol{\omega}_1)$ is closer to the oracle optimal value than $g(\mathbf{x}; \boldsymbol{\omega}_2)$'s.

### 3.1 Surrogate Sharpness

Suppose the oracle lies within the parametric family of the surrogate, there must exist a perturbation neighborhood of the surrogate's parameters that contains the oracle. That is, the oracle can be obtained by adding to the surrogate's parameters a noise vector in this neighborhood. Now, suppose the predictions do not change substantially across the models (including both the oracle and surrogate) in the perturbation neighborhood; the surrogate's predictions (and optimizers) must be close to those of the oracle (see Fig. 1). Motivated by this insight, we consider a potential approach to mitigate the erratic prediction of the surrogate, which is to ensure that its worst-case prediction change across the perturbation neighborhood is sufficiently small. This is formalized below:

$$\mathcal{R}_{\mathcal{X}}(\boldsymbol{\omega}) \triangleq \max_{\|\boldsymbol{\delta}\|_2 \leq \rho} \left| \mathbb{E}_{\mathbf{x} \in \mathcal{X}} \Big[ g(\mathbf{x}; \boldsymbol{\omega} + \boldsymbol{\delta}) \Big] - \mathbb{E}_{\mathbf{x} \in \mathcal{X}} \Big[ g(\mathbf{x}; \boldsymbol{\omega}) \Big] \right|, \tag{4}$$

where $\| \cdot \|_2$ denotes the $\ell_2$-norm, and $\rho > 0$ bounds the maximum norm of the perturbation, which defines the perturbation neighborhood and can be selected via hyper-parameter tuning (see Section 5.1 and Appendix G.4). To ease the notation, we will omit the subscript and use $\|.\|$ to consistently denote the $\ell_2$ norm in the rest of this manuscript. We will also refer to $\mathcal{R}_{\mathcal{X}}(\boldsymbol{\omega})$ in Eq. (4) as the generalized surrogate sharpness, which can be used to regularize surrogate training,

$$\boldsymbol{\omega}_* \triangleq \arg\min_{\boldsymbol{\omega}} \mathcal{L}_{\mathcal{D}}(\boldsymbol{\omega}) \quad \text{s.t.} \quad \mathcal{R}_{\mathcal{X}}(\boldsymbol{\omega}) \leq \epsilon', \tag{5}$$

where $\epsilon'$ is a user-specified threshold. Solving Eq. (5) is non-trivial since $\mathcal{R}_{\mathcal{X}}(\boldsymbol{\omega})$ is neither tractable nor differentiable. To sidestep this, we leverage a theoretical result that will be established later in Section 4, which (informally) states that with high confidence, the generalized surrogate sharpness

$\mathcal{R}_{\mathcal{X}}(\boldsymbol{\omega})$ can be approximated with its empirical estimate $\mathcal{R}_{\mathcal{D}}(\boldsymbol{\omega})$ in Eq. (16),

$$\mathcal{R}_{\mathcal{X}}(\boldsymbol{\omega}) \leq \left(\rho G(\boldsymbol{\omega}) + \frac{1}{2}\rho^2 \lambda_{\max}\right) \cdot \left(\mathcal{R}_{\mathcal{D}}(\boldsymbol{\omega}) + \mathcal{O}\left(\sqrt{\frac{\dim(\boldsymbol{\omega})\log\left(n\|\boldsymbol{\omega}\|^2\right)}{n}}\right)\right), \quad (6)$$

where $G(\boldsymbol{\omega})$ and $\lambda_{\max}$ denote the norm of the expected (parameter) gradient of the surrogate at $\boldsymbol{\omega}$, and the largest eigenvalue of the (parameter) Hessian of the surrogate's expected prediction,

$$G(\boldsymbol{\omega}) \triangleq \left\|\mathop{\mathbb{E}}_{\mathbf{x}\in\mathcal{X}}\left[\nabla_{\boldsymbol{\omega}}g(\mathbf{x};\boldsymbol{\omega})\right]\right\| \quad \text{and} \quad \lambda_{\max} \triangleq \max_{\boldsymbol{\omega}} \lambda_{\max}\left(\nabla_{\boldsymbol{\omega}}^2 \mathop{\mathbb{E}}_{\mathbf{x}\in\mathcal{X}}\left[g(\mathbf{x};\boldsymbol{\omega})\right]\right). \quad (7)$$

When $n$ is sufficiently large, $(\dim(\boldsymbol{\omega})\log(n\|\boldsymbol{\omega}\|^2)/n)^{\frac{1}{2}}$ will be negligibly small. Then, suppose within the search region for $\boldsymbol{\omega}$ (see Assumption 2), $G(\boldsymbol{\omega})$ is bounded by a constant $G$, we can enforce $\mathcal{R}_{\mathcal{X}}(\boldsymbol{\omega}) \leq \epsilon' = (\rho G + (1/2)\rho^2 \lambda_{\max})\epsilon$ via constraining instead $\mathcal{R}_{\mathcal{D}}(\boldsymbol{\omega}) \leq \epsilon$,

$$\boldsymbol{\omega}_* \triangleq \arg\min_{\boldsymbol{\omega}} \mathcal{L}_{\mathcal{D}}(\boldsymbol{\omega}) \quad \text{s.t.} \quad \mathcal{R}_{\mathcal{D}}(\boldsymbol{\omega}) \leq \epsilon. \quad (8)$$

To mitigate the non-differentiability of $\mathcal{R}_{\mathcal{D}}(\boldsymbol{\omega})$, we further propose a practical approximation which reduces $\mathcal{R}_{\mathcal{D}}(\boldsymbol{\omega})$ to a function of the surrogate's gradient norm $\|\nabla_{\boldsymbol{\omega}}g(\mathbf{x};\boldsymbol{\omega})\|$, which is differentiable, allowing the above COP to be solved effectively with existing constrained optimization algorithms. This is discussed in the next section.

**Remark.** Eq. (4) is similar in spirit to the notion of loss sharpness [14] in which the same perturbation model is used to characterize the sharpness or sensitivity of the loss function, $|\mathcal{L}_{\mathcal{D}}(\boldsymbol{\omega}+\boldsymbol{\delta}) - \mathcal{L}_{\mathcal{D}}(\boldsymbol{\omega})|$, to small changes $\|\boldsymbol{\delta}\| \leq \rho$ in the model weights $\boldsymbol{\omega}$. Our work, however, sets a direct focus on the sensitivity or sharpness of the surrogate's prediction – Eq. (4) – which results in a better surrogate regularizer, leading to better empirical performance (see Section 5.3). It also requires a significant and non-trivial adaptation of the theories presented in [14], as detailed later in Section 4.

### 3.2 Practical Algorithms

Let $h(\boldsymbol{\omega}+\boldsymbol{\delta}) \triangleq \mathbb{E}_{\mathbf{x}\in\mathcal{D}}[g(\mathbf{x};\boldsymbol{\omega}+\boldsymbol{\delta})]$ and $h(\boldsymbol{\omega}) \triangleq \mathbb{E}_{\mathbf{x}\in\mathcal{D}}[g(\mathbf{x};\boldsymbol{\omega})]$. The surrogate sharpness can be approximated via the first-order Taylor expansion of $h(\boldsymbol{\omega}+\boldsymbol{\delta})$ at $\boldsymbol{\omega}$:

$$\begin{aligned}
\mathcal{R}_{\mathcal{D}}(\boldsymbol{\omega}) &= \max_{\|\boldsymbol{\delta}\|_2\leq\rho} \left|\mathop{\mathbb{E}}_{\mathbf{x}\in\mathcal{D}}\left[g(\mathbf{x};\boldsymbol{\omega}+\boldsymbol{\delta})\right] - \mathop{\mathbb{E}}_{\mathbf{x}\in\mathcal{D}}\left[g(\mathbf{x};\boldsymbol{\omega})\right]\right| \\
&= \max_{\|\boldsymbol{\delta}\|_2\leq\rho} \left|h(\boldsymbol{\omega}+\boldsymbol{\delta}) - h(\boldsymbol{\omega})\right| \simeq \max_{\|\boldsymbol{\delta}\|_2\leq\rho}\left|\nabla_{\boldsymbol{\omega}}h(\boldsymbol{\omega})^\top\boldsymbol{\delta}\right|,
\end{aligned} \quad (9)$$

Using the Cauchy-Schwartz inequality on the right-hand side of the above and noting that $\boldsymbol{\delta}$ can be selected to make the equality happens,

$$\mathcal{R}_{\mathcal{D}}(\boldsymbol{\omega}) \simeq \max_{\|\boldsymbol{\delta}\|_2\leq\rho}\left|\nabla_{\boldsymbol{\omega}}h(\boldsymbol{\omega})^\top\boldsymbol{\delta}\right| = \max_{\|\boldsymbol{\delta}\|_2\leq\rho}\left\|\nabla_{\boldsymbol{\omega}}h(\boldsymbol{\omega})\right\| \times \left\|\boldsymbol{\delta}\right\| = \rho \times \left\|\nabla_{\boldsymbol{\omega}}h(\boldsymbol{\omega})\right\|. \quad (10)$$

Using the above approximation, the COP in Eq. (8) can be rewritten as

$$\boldsymbol{\omega}_* \triangleq \arg\min_{\boldsymbol{\omega}} \mathcal{L}_{\mathcal{D}}(\boldsymbol{\omega}) \quad \text{s.t.} \quad \rho \cdot \|\nabla_{\boldsymbol{\omega}}h(\boldsymbol{\omega})\| \leq \epsilon. \quad (11)$$

which can be solved via optimizing the corresponding Lagrangian,

$$\boldsymbol{\omega}_* = \arg\min_{\boldsymbol{\omega}} \mathcal{L}(\boldsymbol{\omega},\lambda) \quad \text{where} \quad \mathcal{L}(\boldsymbol{\omega},\lambda) \triangleq \mathcal{L}_{\mathcal{D}}(\boldsymbol{\omega}) + \lambda \cdot \left(\rho \cdot \|\nabla_{\boldsymbol{\omega}}h(\boldsymbol{\omega})\| - \epsilon\right), \quad (12)$$

with $\lambda > 0$ denotes the Lagrange multiplier. This can be solved via our approach below.

**IGNITE.** We can optimize for both $\lambda$ and $\boldsymbol{\omega}$ using the basic differential multiplier method (BDMM) [17], which simultaneously gradient ascent for $\lambda$ and gradient descent for $\boldsymbol{\omega}$, resulting in the following update rules:

$$\boldsymbol{\omega}^{t+1} = \boldsymbol{\omega}^t - \eta_{\boldsymbol{\omega}} \cdot \left(\nabla_{\boldsymbol{\omega}}\mathcal{L}_{\mathcal{D}}(\boldsymbol{\omega}^t) + \lambda^t \cdot \rho \cdot \nabla_{\boldsymbol{\omega}}\left\|\nabla_{\boldsymbol{\omega}}h(\boldsymbol{\omega}^t)\right\|\right), \quad (13)$$

$$\lambda^{t+1} = \lambda^t + \eta_\lambda \cdot \left(\rho \cdot \left\|\nabla_{\boldsymbol{\omega}}h\left(\boldsymbol{\omega}^t\right)\right\| - \epsilon\right), \quad (14)$$

**Algorithm 1 IGNITE**

---

**Input:** offline data $\mathcal{D} = \{(\mathbf{x}_i, z_i)\}_{i=1}^n$; initial surrogate $g(\mathbf{x}; \boldsymbol{\omega}^{(0)})$; no. of iterations $T$; batch size $m$; Lagrange multiplier $\lambda$; perturbation radius $\rho$ and scalar $r$; step sizes $\eta_{\boldsymbol{\omega}}$ and $\eta_\lambda$; threshold $\epsilon$.

1: Initialize $\boldsymbol{\omega}^{(1)} \leftarrow \boldsymbol{\omega}^{(0)}$ and $\lambda^{(1)} \leftarrow \lambda$     7:    Compute $\hat{\boldsymbol{\omega}} = \boldsymbol{\omega}^{(t)} + r \cdot g_2 / \|g_2\|$
2: **for** $t \leftarrow 1 : T$ **do**     8:    Compute $g_3 = m^{-1} \sum_{i=1}^m \nabla_{\boldsymbol{\omega}} g(\mathbf{x}_i; \hat{\boldsymbol{\omega}})$
3:     Sample $\mathcal{B} = \{(\mathbf{x}_i, z_i)\}_{i=1}^m \sim \mathcal{D}$     9:    Compute $g^{(t)} = g_1 + \lambda^{(t)} \rho r^{-1} (g_3 - g_2)$
4:     Compute $\hat{z}_i = g(\mathbf{x}_i; \boldsymbol{\omega}^{(t)})$ for $i \in [m]$     10:    Update $\boldsymbol{\omega}^{(t+1)} \leftarrow \boldsymbol{\omega}^{(t)} - \eta_{\boldsymbol{\omega}} g^{(t)}$
5:     Compute $g_1 = m^{-1} \sum_{i=1}^m \nabla_{\boldsymbol{\omega}} \ell(\hat{z}_i, z_i)$     11:    Update $\lambda^{(t+1)} \leftarrow \lambda^{(t)} + \eta_\lambda (\rho \|g_2\| - \epsilon)$
6:     Compute $g_2 = m^{-1} \sum_{i=1}^m \nabla_{\boldsymbol{\omega}} \hat{z}_i$     12: **end for**
    13: **return** learned surrogate $\boldsymbol{\omega}^{(T+1)}$

---

where $\boldsymbol{\omega}^t$ represents the surrogate's parameter estimate at iteration $t$, $\lambda^t$ is the Lagrange multiplier estimate at iteration $t$, $\eta_{\boldsymbol{\omega}}$ is the step size for updating $\boldsymbol{\omega}$, and $\eta_\lambda$ is the step size for updating $\lambda$. We name this method **IGNITE**. We also conduct grid search to select the optimal value for $\rho$, as mentioned in Section 5.1.

**Remark.** As an alternative approach, we can also treat $\lambda$ as a hyper-parameter and optimize for $\boldsymbol{\omega}$ using gradient descent; we call this method **IGNITE-2** . The detailed algorithms, hyper-parameter selection, and experimental results of **IGNITE-2** are reported in Appendix G.2.

Both of the above methods require differentiating $\|\nabla_{\boldsymbol{\omega}} h(\boldsymbol{\omega})\|$ with respect to $\boldsymbol{\omega}$, which involves computing the expensive Hessian of $h(\boldsymbol{\omega})$. Fortunately, this expensive computation can be avoided by using the gradient approximation technique detailed in Appendix F. To summarize, we provide a complete pseudo-code of **IGNITE** in Algorithm 1 whose steps 3-8 implement the approximation in Eq. (84) and Eq. (85) of Appendix F.

## 4 Theoretical Analysis

In this section, we will provide a detailed theoretical analysis to substantiate our earlier (informal) statement in Eq. (6) that with high confidence, reducing the empirical sharpness $\mathcal{R}_{\mathcal{D}}(\boldsymbol{\omega})$ on the offline data will also reduce its generalized sharpness $\mathcal{R}_{\mathcal{X}}(\boldsymbol{\omega})$. We will show that this is true (see Theorem 1) under certain choices and mild assumptions of the surrogate model (see Assumptions 1 and 2).

**Assumption 1.** The output of the above surrogate function $g(\mathbf{x}; \boldsymbol{\omega})$ is bounded within $[0, 1]$.

**Assumption 2.** $\lambda_{\min}\left(\nabla_{\boldsymbol{\omega}}^2 \mathbb{E}[g(\mathbf{x}; \boldsymbol{\omega})]\right) > 0$ for all $\|\boldsymbol{\omega}\| \leq \tau$ for some $\tau > 0$.

We note that the specific bound within $[0, 1]$ in Assumption 1 is meant to ease the technical presentation of our theoretical analysis. Otherwise, it can be extended straightforwardly to any bounded functions. Furthermore, we also show below that it is indeed possible to find a non-trivial surrogate function that is bounded and satisfies Assumption 2.

**Theorem 1.** *There exists $\tau > 0$ and $\boldsymbol{\omega}_+$, and a non-linear function $r(\mathbf{x}; \boldsymbol{\omega})$ such that,*

$$g(\mathbf{x}; \boldsymbol{\omega}) \triangleq r(\mathbf{x}; \boldsymbol{\omega}_+) + (\boldsymbol{\omega} - \boldsymbol{\omega}_+)^\top \nabla_{\boldsymbol{\omega}} r(\mathbf{x}; \boldsymbol{\omega}_+) + \frac{1}{2}(\boldsymbol{\omega} - \boldsymbol{\omega}_+)^\top \nabla_{\boldsymbol{\omega}}^2 r(\mathbf{x}; \boldsymbol{\omega}_+)(\boldsymbol{\omega} - \boldsymbol{\omega}_+) \quad (15)$$

*satisfies Assumption 2 and is bounded on $\{\boldsymbol{\omega} \mid \|\boldsymbol{\omega}\| \leq \tau\}$. Detailed derivation of this theorem is deferred to Appendix A.*

For such surrogate functions, their generalized sharpness can be upper-bound by a function of their empirical sharpness on the offline data. As detailed below, the bound depends on both the size of the surrogate $\dim(\boldsymbol{\omega})$ and the number $n$ of offline data points.

**Theorem 2.** *For any $\rho > 0$, $m = \dim(\boldsymbol{\omega})$ and $2/(m\lambda_{\min}) \geq \sigma^2 > 0$ with $\lambda_{\min}$ being defined in Assumption 2, the following holds simultaneously for all $g(.; \boldsymbol{\omega})$ for which Assumption 2 is met,*

$$\mathcal{R}_{\mathcal{X}}(\boldsymbol{\omega}) \leq \frac{1}{\sigma^2 m \lambda_{\min}} \left( 2G(\boldsymbol{\omega})\rho + \lambda_{\max}\rho^2 \right)$$

$$\times \left( \mathcal{R}_{\mathcal{D}}(\boldsymbol{\omega}) + \frac{1}{\sqrt{n-1}} \left( m \log \left( 1 + \frac{\|\boldsymbol{\omega}\|^2}{m\sigma^2} \left( 1 + \sqrt{\frac{\log n}{m}} \right)^2 \right) + P(n, m, \alpha) \right)^{\frac{1}{2}} \right) \quad (16)$$

*with probability at least $1 - \alpha$ over the random choice of the offline dataset, and with $P(n, m, \alpha) = 2\log(n/\alpha) + 4\log(8n + 4m)$. Detailed derivation of this theorem is deferred to Appendix E.*

**Proof Sketch.** For clarity, we will provide below a proof sketch of Theorem 2, which highlights the key steps in our derivation. Due to limited space, the specific of each step is deferred to the Appendix E. First, we note that

$$\mathcal{R}_{\mathcal{X}}(\boldsymbol{\omega}) = \max_{\|\boldsymbol{\delta}\| \leq \rho} \left\{ F_{\mathcal{X}}(\boldsymbol{\omega} + \boldsymbol{\delta}) \triangleq \left| \mathbb{E}_{\mathcal{X}}[g(\mathbf{x}; \boldsymbol{\omega} + \boldsymbol{\delta})] - \mathbb{E}_{\mathcal{X}}[g(\mathbf{x}; \boldsymbol{\omega})] \right| \right\}, \quad (17)$$

$$\mathcal{R}_{\mathcal{D}}(\boldsymbol{\omega}) = \max_{\|\boldsymbol{\delta}\| \leq \rho} \left\{ F_{\mathcal{D}}(\boldsymbol{\omega} + \boldsymbol{\delta}) \triangleq \left| \mathbb{E}_{\mathcal{D}}[g(\mathbf{x}; \boldsymbol{\omega} + \boldsymbol{\delta})] - \mathbb{E}_{\mathcal{D}}[g(\mathbf{x}; \boldsymbol{\omega})] \right| \right\}. \quad (18)$$

A relation between $\mathcal{R}_{\mathcal{X}}(\boldsymbol{\omega})$ and $\mathcal{R}_{\mathcal{D}}(\boldsymbol{\omega})$ can then be derived in three steps:

**1.** Upper-bound $\mathbb{E}_{\boldsymbol{\delta} \sim \mathbb{N}(0,\sigma^2\mathbf{I})}[F_{\mathcal{X}}(\boldsymbol{\omega} + \boldsymbol{\delta})]$ with a function of $\mathbb{E}_{\boldsymbol{\delta} \sim \mathbb{N}(0,\sigma^2\mathbf{I})}[F_{\mathcal{D}}(\boldsymbol{\omega} + \boldsymbol{\delta})]$. This can be achieved via a direct application of the PAC-Bayes bound [18] which views the perturbed model $\boldsymbol{\omega} + \boldsymbol{\delta}$ as a random hypothesis sampled from the posterior $\mathbb{N}(\boldsymbol{\omega}, \sigma^2\mathbf{I})$ – see Appendix B.

**2.** Upper-bound $\mathbb{E}_{\boldsymbol{\delta} \sim \mathbb{N}(0,\sigma^2\mathbf{I})}[F_{\mathcal{D}}(\boldsymbol{\omega} + \boldsymbol{\delta})]$ with a function of $\mathcal{R}_{\mathcal{D}}(\boldsymbol{\omega})$. This can be achieved using a similar proving technique adopted from [14] – see Appendix C.

**3.** Find $\xi > 0$ such that $\mathcal{R}_{\mathbf{X}}(\boldsymbol{\omega})$ can be upper-bounded with $\xi \cdot \mathbb{E}_{\boldsymbol{\delta} \sim \mathbb{N}(0,\sigma^2\mathbf{I})}[F_{\mathcal{X}}(\boldsymbol{\omega} + \boldsymbol{\delta})]$ where $\xi$ is a small scaling factor. To achieve this, we will find the lower-bound for $\mathbb{E}_{\boldsymbol{\delta} \sim \mathbb{N}(0,\sigma^2\mathbf{I})}[F_{\mathcal{X}}(\boldsymbol{\omega} + \boldsymbol{\delta})]$ using the Taylor remainder theorem to expand it around $\boldsymbol{\omega}$, which can be lower-bounded using the minimum eigenvalue of its Hessian at $\boldsymbol{\omega}$. Using the same approach, we can upper-bound $\mathcal{R}_{\mathbf{X}}(\boldsymbol{\omega})$ with the maximum eigenvalue of the same Hessian – see Appendix D.

Finally, we set $\xi$ so that the upper-bound of $\mathcal{R}_{\mathbf{X}}(\boldsymbol{\omega})$ is smaller than the multiplication of $\xi$ with the lower-bound of $\mathbb{E}[F_{\mathcal{X}}(\boldsymbol{\omega} + \boldsymbol{\delta})]$. To tighten the bound, we choose the smallest possible value of $\xi$ such that the bound still holds. Lining up the results of the above steps then shows that $\mathcal{R}_{\mathcal{X}}(\boldsymbol{\omega})$ can then be bounded with a function of $\xi \cdot \mathcal{R}_{\mathcal{D}}(\boldsymbol{\omega})$. See Appendix E for a complete derivation.

**Remark.** Note that Theorem 2 is more general than its informal statement in Eq. (6), which can be reproduced by choosing $\sigma^2 = 2/(m\lambda_{\min})$ in Eq. (16) to make the bound tightest.

## 5 Experiments

This section evaluates the efficacy of our proposed method **IGNITE** in improving state-of-the-art offline optimizers. We describe our experiment settings in Section 5.1 and report detailed empirical results in Section 5.2. We also provide additional ablation studies of our method in Section 5.3.

### 5.1 Benchmarks, Baselines, and Evaluation

**Benchmark Tasks.** Our explorations focus on four real-world tasks from Design-Bench[3] [4], covering both discrete (**TF-Bind-8** and **TF-Bind-10**) and continuous domains (**Ant Morphology** [19] and **D'Kitty Morphology** [20]).

**Baselines.** We meticulously curated a diverse set of 11 widely acknowledged offline optimizers for comparative analysis. This ensemble comprises **BO-qEI** [4], **CbAS** [1], **RoMA** [3], **ICT** [15],

---

[3]We omit domains marked for their high inaccuracy and noise in oracle functions from prior works (**ChEMBL**, **Hopper**, and **Superconductor**), as well as those deemed excessively expensive to evaluate (**NAS**).

Table 1: The percentage improvement in performance achieved by **IGNITE** across all tasks and baseline algorithms at the **100th percentile** level is presented. **Gain** signifies the percentage gain over the baseline performance (Base).

| Algorithms | | Ant Morphology | | D'Kitty Morphology | | TF Bind 8 | | TF Bind 10 | |
|---|---|---|---|---|---|---|---|---|---|
| | | Performance | Gain | Performance | Gain | Performance | Gain | Performance | Gain |
| $\mathfrak{D}$(**best**) | | 0.565 | | 0.884 | | 0.565 | | 0.884 | |
| **REINF-ORCE** | Base | 0.255 ± 0.036 | | 0.546 ± 0.208 | | 0.929 ± 0.043 | | 0.635 ± 0.028 | |
| | IGNITE | 0.282 ± 0.021 | +2.7% | 0.642 ± 0.160 | +9.6% | 0.944 ± 0.030 | +1.5% | 0.670 ± 0.060 | +3.5% |
| **GA** | Base | 0.303 ± 0.027 | | 0.881 ± 0.016 | | 0.980 ± 0.016 | | 0.651 ± 0.033 | |
| | IGNITE | 0.320 ± 0.044 | +1.7% | 0.886 ± 0.017 | +0.5% | 0.985 ± 0.010 | +0.5% | 0.653 ± 0.043 | +0.2% |
| **ENS-MEAN** | Base | 0.376 ± 0.060 | | 0.888 ± 0.010 | | 0.985 ± 0.009 | | 0.649 ± 0.036 | |
| | IGNITE | 0.435 ± 0.058 | +5.9% | 0.896 ± 0.013 | +0.8% | 0.987 ± 0.007 | +0.2% | 0.662 ± 0.091 | +1.3% |
| **ENS-MIN** | Base | 0.385 ± 0.067 | | 0.889 ± 0.014 | | 0.980 ± 0.012 | | 0.681 ± 0.095 | |
| | IGNITE | 0.468 ± 0.062 | +8.3% | 0.897 ± 0.010 | +0.8% | 0.986 ± 0.010 | +0.6% | 0.705 ± 0.118 | +2.4% |
| **CbAS** | Base | 0.854 ± 0.042 | | 0.895 ± 0.012 | | 0.919 ± 0.044 | | 0.635 ± 0.041 | |
| | IGNITE | 0.859 ± 0.039 | +0.5% | 0.900 ± 0.015 | +0.5% | 0.921 ± 0.042 | +0.2% | 0.652 ± 0.055 | +1.7% |
| **MINs** | Base | 0.905 ± 0.023 | | 0.944 ± 0.008 | | 0.892 ± 0.046 | | 0.643 ± 0.062 | |
| | IGNITE | 0.911 ± 0.024 | +0.6% | 0.945 ± 0.007 | +0.1% | 0.930 ± 0.041 | +3.8% | 0.647 ± 0.058 | +0.4% |
| **RoMA** | Base | 0.569 ± 0.086 | | 0.821 ± 0.019 | | 0.665 ± 0.000 | | 0.550 ± 0.008 | |
| | IGNITE | 0.615 ± 0.085 | +4.6% | 0.834 ± 0.012 | +1.3% | 0.665 ± 0.000 | +0.0% | 0.553 ± 0.000 | +0.3% |
| **COMs** | Base | 0.897 ± 0.031 | | 0.931 ± 0.013 | | 0.955 ± 0.030 | | 0.645 ± 0.038 | |
| | IGNITE | 0.901 ± 0.030 | +0.4% | 0.934 ± 0.010 | +0.3% | 0.952 ± 0.043 | -0.3% | 0.638 ± 0.053 | -0.7% |
| **CMA-ES** | Base | 1.955 ± 1.484 | | 0.724 ± 0.002 | | 0.928 ± 0.040 | | 0.668 ± 0.035 | |
| | IGNITE | 1.957 ± 1.910 | +0.2% | 0.724 ± 0.001 | +0.0% | 0.927 ± 0.043 | -0.1% | 0.673 ± 0.044 | +0.5% |
| **BO-qEI** | Base | 0.812 ± 0.000 | | 0.896 ± 0.000 | | 0.787 ± 0.112 | | 0.628 ± 0.000 | |
| | IGNITE | 0.812 ± 0.000 | +0.0% | 0.896 ± 0.000 | +0.0% | 0.843 ± 0.109 | +5.6% | 0.628 ± 0.000 | +0.0% |
| **ICT** | Base | 0.937 ± 0.023 | | 0.946 ± 0.014 | | 0.892 ± 0.055 | | 0.647 ± 0.025 | |
| | IGNITE | 0.935 ± 0.032 | -0.2% | 0.962 ± 0.018 | +1.6% | 0.923 ± 0.038 | +3.1% | 0.652 ± 0.074 | +0.5% |

**CMA-ES** [21], **COMs** [13], **MINs** [12], **REINFORCE** [22], and three variations of gradient ascent (**GA**, **ENS-MIN**, **ENS-MEAN**), corresponding to vanilla gradient ascent, the min ensemble of gradient ascent, and the mean ensemble of gradient ascent, respectively.

**Evaluation Protocol.** To ensure a comprehensive assessment, we follow the methodology in [4]. Each method generates 128 optimized design candidates evaluated by the oracle function. The candidates' performances are ranked, and the 50-th, 75-th, and 100-th percentile levels are recorded. All results are averaged over 16 independent runs to ensure reliability.

**Hyper-parameter Configuration.** For each baseline algorithm, we carefully configure optimal hyper-parameters as outlined in [4]. Our method **IGNITE** introduces five additional hyper-parameters: $\lambda$, $\rho$, $r$, $\eta_\lambda$, and $\epsilon$. The hyper-parameter $\lambda$, an initial value for the regularizer coefficient, is set to $0.01$ through a grid search within $\{0.0001, 0.001, 0.01\}$. The hyper-parameters $\rho$ and $r$ are chosen from $\{0.01, 0.05, 0.1, 0.2\}$, with $\rho$ set to $0.05$ and $r$ set to $0.05$ for **IGNITE**. Additionally, **IGNITE** uses $\eta_\lambda = 1e-3$ and $\epsilon = 0.1$, which are determined via the experiments in Section 5.3.

## 5.2 Results and Discussion

In this section, we presented the percentage improvement over baseline performance attained by **IGNITE** when it is applied to existing baselines. We have evaluated this at the 50-th, 80-th, and 100-th percentile levels. However, due to limited space, we only report results of the 100-th percentile level in the main text. The other results are instead deferred to Appendix G.3.

**Results on Continuous Tasks:** The first two columns of Table 1 show that out of 22 cases involving 11 baseline algorithms across 2 tasks, the **IGNITE** regularizer enhances baseline performance in 18 cases, with improvements reaching up to 9.6%. In only 1 out of 22 instances **IGNITE** slightly decreases performance by 0.2%, which is negligible. Even in cases where performance is not improved, **IGNITE** reduces the variance in results from 0.2% to 0.1% for the **CMA-ES** baseline on the **D'Kitty Morphology** task. Additionally, **IGNITE** helps establish new state-of-the-art (SOTA)

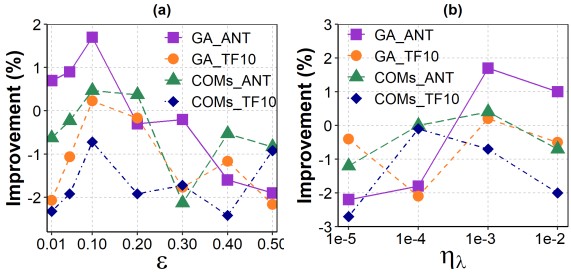

Figure 2: The percentage improvement in performance achieved by **IGNITE** across different algorithms (**COMS** and **GA**) and tasks (**ANT** and **TF10**) in the changes of (a) threshold $\epsilon$ and (b) step size $\eta_\lambda$.

Table 2: Percentage improvement over the baseline of **IGNITE**, SAM [14], and L1, L2 regularization across all tasks.

| Algorithms | | Ant | D'Kitty | TF Bind 8 | TF Bind 10 |
|---|---|---|---|---|---|
| **REINF-ORCE** | **IGNITE** | **2.7%** | **9.6%** | **1.5%** | **3.5%** |
| | SAM | 1.1% | 7.9% | 1.1% | 0.2% |
| | L1-Reg. | 1.0% | 5.2% | 1.0% | 0.3% |
| | L2-Reg. | 1.0% | 4.2% | 0.9% | 0.1% |
| **GA** | **IGNITE** | **1.7%** | **0.5%** | **0.5%** | 0.2% |
| | SAM | 0.7% | -1.3% | 0.2% | **1.1%** |
| | L1-Reg. | 1.0% | 0.0% | 0.1% | -0.8% |
| | L2-Reg. | 1.1% | -0.7% | 0.2% | -0.4% |

performances in both tasks. For example, in the **Ant Morphology** task, it raises the SOTA baseline **CMA-ES** from $195.5\%$ to $195.7\%$. In the **D'Kitty Morphology** task, **IGNITE** achieves a new SOTA of $96.2\%$ with the **ICT** baseline.

**Results on Discrete Tasks:** The last two columns of Table 1 show the impact of the **IGNITE** regularizer on the performance of baseline algorithms in two discrete domains (**TF-BIND-8** and **TF-BIND-10**). Similar to the continuous tasks, **IGNITE** significantly enhances baseline performance in most cases (17 out of 22), with improvements of up to $5.6\%$. There are only 3 instances where integrating **IGNITE** results in a minor performance decrease of up to $0.7\%$, which is negligible. Additionally, in certain instances, **IGNITE** not only improves baseline performance but also establishes new state-of-the-art (SOTA) results. For example, on **TF-BIND-8** and **TF-BIND-10**, the original SOTA performances of $98.5\%$ and $68.1\%$ achieved by **ENS-MEAN** and **ENS-MIN**, respectively, are elevated to $98.7\%$ and $70.5\%$ with the addition of **IGNITE** , setting new SOTA records.

In summary, **IGNITE** consistently maintains a high probability of $91\%$ (40 out of 44) of not degrading baseline performance. There is a high likelihood ($79.55\% = 35$ out of 44 cases) of improving baseline performance, with an average improvement of approximately $1.91\%$ and a notable peak improvement of $9.6\%$. Conversely, **IGNITE** also exhibits a relatively low probability ($9.09\% = 4$ out of 44 cases) of decreasing performance, with an average degradation of approximately $0.3\%$ and a minor peak degradation of $0.7\%$.Additionally, there is a minor probability ($11.36\% = 5$ out of 44 cases) of maintaining baseline performance.

## 5.3 Ablation Experiments

In this section, we conduct additional experiments to assess the sensitivity of two representative baselines, **COMs** and **GA**, when regularized with **IGNITE**, to variations in hyper-parameters $\epsilon$ and $\eta_\lambda$. Additionally, we perform experiments to compare the efficacy of **IGNITE** with other commonly used regularization methods.

**Changing threshold $\epsilon$.** We assess the performance enhancement of **COMs** and **GA** when regularized with **IGNITE** using various values of $\epsilon$ from the set $\{0.01, 0.05, 0.1, 0.2, 0.3, 0.4, 0.5\}$. A high $\epsilon$ value may result in a surrogate that is overly sharp, potentially hampering the search process and hindering the discovery of optimal designs. Figure 2(a) demonstrates that excessively high $\epsilon$ values lead to negative improvements. Conversely, excessively low $\epsilon$ values may cause the regularizer to dominate the original loss, resulting in a surrogate that is not well-fitted to the offline data. Negative improvements are observed with $\epsilon = 0.01$ and $0.05$ in Figure 2(a). As a result, we determine $\epsilon = 0.1$ as the optimal value based on these observations.

**Changing step size $\eta_\lambda$.** The step size $\eta_\lambda$ controls the rate at which $\lambda$ is updated during the optimization process. It's essential to choose an appropriate step size, avoiding it being either too large or too small. Figure 2(b) demonstrates that an $\eta_\lambda$ value of $1e-3$ yields optimal results.

Table 3: Comparison of the surrogate sharpness — approximately $\rho\|\nabla_\omega h(\omega)\|$ in Eq. 10 — with and without IGNITE. These surrogate sharpness values were computed on unseen data, which are design candidates found by the **GA** and **REINFORCE** before and after being equipped with **IGNITE**.

| Algorithms | Ant | TF Bind 10 |
|---|---|---|
| **REINFORCE** | 0.18 | 0.24 |
| **REINFORCE + IGNITE** | 0.09 | 0.14 |
| **GA** | 1.88 | 1.07 |
| **GA + IGNITE** | 1.69 | 0.63 |

Table 4: The percentage improvement in performance achieved by **IGNITE** at the 100th percentile level for **Superconductor** and **Chembl** tasks.

| | Algorithms | Performance |
|---|---|---|
| **Super-conductor** | **REINFORCE** | 0.471 ± 0.011 |
| | **REINFORCE + IGNITE** | 0.492 ± 0.015 (+2.1%) |
| | **GA** | 0.514 ± 0.021 |
| | **GA + IGNITE** | 0.517 ± 0.011 (+0.3%) |
| **Chembl** | **REINFORCE** | 0.634 ± 0.001 |
| | **REINFORCE + IGNITE** | 0.636 ± 0.008 (+0.2%) |
| | **GA** | 0.635 ± 0.005 |
| | **GA + IGNITE** | 0.640 ± 0.009 (+0.5%) |

**Comparing IGNITE with other regularization methods.** We conduct a comparative experiment to assess the performance improvement achieved by using the **IGNITE** regularizer compared to other regularizers, including L1, L2, and SAM [14] (where SAM is considered as a loss sharpness regularization with a coefficient equal to 1). Table 2 presents the results obtained on two baseline algorithms, **REINFORCE** and **GA**, across four tasks. Overall, **IGNITE** outperforms the other regularizers in most cases, with the largest difference compared to the second best being 3.2% with **REINFORCE** on the **TF-BIND-10** task. Additionally, **IGNITE** achieves positive improvements in all cases. Conversely, while the simple regularizers L1 and L2 help boost performance with **REINFORCE**, they lead to performance degradation with **GA**. SAM outperforms **IGNITE** with the **GA** baseline on the **TF-BIND-10** task and shows notable improvement with **REINFORCE** on the **D'Kitty** task, though its integration with **GA** leads to a performance drop. Furthermore, we show that **IGNITE** also outperforms SAM when integrating with **CbAS** and **BO-qEI** in Appendix H.

**Surrogate sharpness on unseen data before and after using IGNITE.** We also conduct an experiment measuring the sharpness of the surrogate model, approximated by $\rho\|\nabla_\omega h(\omega)\|$ as defined in Eq. (10), with and without **IGNITE**. This sharpness is computed on unseen data points which are, specifically, the design candidates generated by the **GA** and **REINFORCE** baselines. By testing on these unseen candidates, we simulate the out-of-distribution (OOD) conditions that are critical in assessing generalization in optimization tasks. Table 3 reports these surrogate sharpness measurements for some baselines with and without using the **IGNITE** regularizer. The results demonstrate consistently that the **IGNITE** regularizer helps reduce the surrogate sharpness on unseen data, as anticipated. This reduction indicates that **IGNITE** is effective in smoothing the surrogate model's landscape, leading to better and more stable generalized performance.

**Results on tasks with noisy data.** To demonstrate **IGNITE**'s robust performance in scenarios with noisy oracle, we conducted additional experiments using the **GA** and **REINFORCE** baselines on two benchmark tasks with particularly noisy oracles: **Superconductor** and **Chembl**. For each baseline, we compared its achieved performance with and without the regularizing effect of **IGNITE**, as shown in Table 4. Across both tasks, **IGNITE** helps improve the baseline performance substantially, achieving up to a 2.1% increase for **REINFORCE** on the **Superconductor** task, highlighting **IGNITE**'s ability to enhance performance robustness even in settings with noisy oracles.

## 6 Conclusion

This paper introduces the concept of generalized surrogate sharpness in offline optimization, resulting in the development of a new regularization technique, **IGNITE**. Our theoretical analysis demonstrates that reducing surrogate sharpness on an offline dataset provably decreases its generalized sharpness on unseen data. Empirically, **IGNITE** consistently maintains a high probability (91%) of not degrading baseline performance and a 79.55% likelihood of improving it, with a peak improvement of 9.6%. Additionally, we believe that our novel technique can be adapted to related domains such as robust optimization (RO) and reinforcement learning (RL), suggesting potential future research directions.

## Acknowledgments and Disclosure of Funding

This work was funded by Vingroup Joint Stock Company (Vingroup JSC),Vingroup, and supported by Vingroup Innovation Foundation (VINIF) under project code VINIF.2021.DA00128.

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

# Proofs of Theoretical Results and Additional Experimental Results

## A  Proof of Theorem 1

**Part I. Minimum Eigenvalue of Parameter Hessian is Positive.**

First, we re-state the choice of our surrogate in Theorem 1,

$$g(\mathbf{x};\boldsymbol{\omega}) \triangleq r(\mathbf{x};\boldsymbol{\omega}_+) + (\boldsymbol{\omega} - \boldsymbol{\omega}_+)^\top \nabla_{\boldsymbol{\omega}} r(\mathbf{x};\boldsymbol{\omega}_+) + \frac{1}{2}(\boldsymbol{\omega} - \boldsymbol{\omega}_+)^\top \nabla_{\boldsymbol{\omega}}^2 r(\mathbf{x};\boldsymbol{\omega}_+)(\boldsymbol{\omega} - \boldsymbol{\omega}_+) \quad (19)$$

Regardless of the choice of $r(\mathbf{x};\boldsymbol{\omega})$, differentiating $g(\mathbf{x};\boldsymbol{\omega})$ with respect to $\boldsymbol{\omega}$ yields,

$$\nabla_{\boldsymbol{\omega}}^2 g(\mathbf{x};\boldsymbol{\omega}) = \nabla_{\boldsymbol{\omega}}^2 r(\mathbf{x};\boldsymbol{\omega}_+). \quad (20)$$

which implies the parameter Hessian of $\nabla_{\boldsymbol{\omega}}^2 g(\mathbf{x};\boldsymbol{\omega})$ is always a constant matrix depending on the specific choice of $r(\mathbf{x};\boldsymbol{\omega})$ and a reference point $\boldsymbol{\omega}_+$. Hence, it follows trivially that,

$$\lambda_{\min}\left(\nabla_{\boldsymbol{\omega}}^2 g(\mathbf{x};\boldsymbol{\omega})\right) = \lambda_{\min}\left(\nabla_{\boldsymbol{\omega}}^2 r(\mathbf{x};\boldsymbol{\omega}_+)\right). \quad (21)$$

Therefore, if we can find $r(\mathbf{x};\boldsymbol{\omega})$ such that its parameter Hessian is strictly positive definite at $\boldsymbol{\omega}_+$, the parameter Hessian of $g(\mathbf{x};\boldsymbol{\omega})$ will always be strictly positive definite regardless of $\boldsymbol{\omega}$. This means $g(\mathbf{x};\boldsymbol{\omega})$ will be $\lambda_{\min}$-strongly convex in $\boldsymbol{\omega}$ with $\lambda_{\min} > 0$. As a result, its expectation $\mathbb{E}[g(\mathbf{x};\boldsymbol{\omega})]$ will also be $\lambda_{\min}$-strongly convex with $\lambda_{\min} > 0$ or equivalently, Assumption 2 holds.

This can be done by choosing $r(\mathbf{x};\boldsymbol{\omega})$ to be any quantile prediction of a linear regressor with a (pre-trained) random feature map. For example,

$$r(\mathbf{x};\boldsymbol{\omega}) \triangleq \mathbb{E}_{\boldsymbol{\psi}}\left[\boldsymbol{\omega}^\top \boldsymbol{\psi}(\mathbf{x})\right] + \frac{\gamma}{2}\mathbb{V}_{\boldsymbol{\psi}}[\boldsymbol{\omega}^\top \boldsymbol{\psi}(\mathbf{x})]^{\frac{1}{2}} \quad (22)$$

$$\text{where} \quad \boldsymbol{\psi}(\mathbf{x}) \sim \mathbb{N}\left(\mathbf{m}(\mathbf{x}), \mathrm{diag}[\mathbf{v}(\mathbf{x})]\right), \quad (23)$$

where $\mathbb{V}[.]$ denote the variance function. We can interpret the random feature $\boldsymbol{\psi}(\mathbf{x})$ is sampled from a pre-trained VAE [23] with two deep neural nets characterizing the mean and variance functions, $\mathbf{m}(\mathbf{x})$ and $\mathbf{v}(\mathbf{x})$. These functions can be pre-trained and post-processed in whatever ways we need to control their value ranges. Their parameters are then frozen when we fit $r(\mathbf{x};\boldsymbol{\omega})$ with respect to $\boldsymbol{\omega}$.

To understand the behavior of $r(\mathbf{x};\boldsymbol{\omega})$, we first note below the closed-form expression of Eq. (23),

$$r(\mathbf{x};\boldsymbol{\omega}) = \boldsymbol{\omega}^\top \mathbf{m}(\mathbf{x}) + \frac{\gamma}{2}\left(\boldsymbol{\omega}^\top \mathbf{A}\boldsymbol{\omega}\right)^{\frac{1}{2}}. \quad (24)$$

where $\mathbf{A} \triangleq \mathrm{diag}[\mathbf{v}(\mathbf{x})]$. Differentiating twice both sides of Eq. (24) at $\boldsymbol{\omega}_+$, we have

$$\nabla_{\boldsymbol{\omega}}^2 r(\mathbf{x};\boldsymbol{\omega}_+) \triangleq \frac{\gamma}{2}\cdot\left(\boldsymbol{\omega}_+^\top \mathbf{A}\boldsymbol{\omega}_+\right)^{-\frac{3}{2}}\mathbf{A}\left(\left(\boldsymbol{\omega}_+^\top \mathbf{A}\boldsymbol{\omega}_+\right)\mathbf{I} - \boldsymbol{\omega}_+\boldsymbol{\omega}_+^\top \mathbf{A}\right). \quad (25)$$

We can now choose $\boldsymbol{\omega}_+ = (1/m)\sum_{i=1}^m \mathbf{e}_i$ where $\mathbf{e}_1, \mathbf{e}_2, \ldots, \mathbf{e}_m$ are the eigenvectors of $\mathbf{A}$. Since $\mathbf{A}$ is diagonal, these are also the one-hot vectors and their corresponding eigenvalues are the entries on the diagonal of $\mathbf{A}$ – i.e., $\lambda_i(\mathbf{A}) = [\mathbf{A}]_{ii}$. With this choice, a closed-form expression of $\nabla_{\boldsymbol{\omega}}^2 r(\mathbf{x};\boldsymbol{\omega}_+)$ in terms of the entries of $\mathbf{A}$ can be derived as follow,

$$\nabla_{\boldsymbol{\omega}}^2 r(\mathbf{x};\boldsymbol{\omega}_+) \triangleq \frac{\gamma}{2}\cdot\left(\frac{1}{m^2}\sum_{i=1}^m [\mathbf{A}]_{ii}\right)^{-\frac{3}{2}}\mathbf{A}\mathbf{B}, \quad (26)$$

where $\mathbf{B}$ is another diagonal matrix with entries $[\mathbf{B}]_{ii} = m^{-2}(\sum_{\iota=1}^m [\mathbf{A}]_{\iota\iota} - [\mathbf{A}]_{ii})$. As such, it is trivial to see that if we choose the activation unit (i.e., sigmoid or ReLU) for $\mathbf{v}(\mathbf{x})$ such that its output is positive, then the entries of both $\mathbf{A}$ and $\mathbf{B}$ are positive.

Finally, note that Eq. (26) is a scaled matrix product of two diagonal matrices, $\mathbf{A}$ and $\mathbf{B}$, which will result in a diagonal matrix. Its entries will be positive if the entries of $\mathbf{A}$ and $\mathbf{B}$ are positive, as enforced above. This means the minimum eigenvalue of $\nabla_{\boldsymbol{\omega}}^2 r(\mathbf{x};\boldsymbol{\omega}_+)$ is positive as desired. □

**Part II. Boundedness on $\{\mathbf{w} : \|\mathbf{w}\|^2 \leq \tau\}$.**

Rearranging terms in the expression of $g(\mathbf{x}; \boldsymbol{\omega})$ in Eq. (19),

$$g(\mathbf{x}; \boldsymbol{\omega}) \triangleq \boldsymbol{\omega}^\top \left( \nabla_{\boldsymbol{\omega}} r(\mathbf{x}; \boldsymbol{\omega}_+) - \nabla_{\boldsymbol{\omega}}^2 r(\mathbf{x}; \boldsymbol{\omega}_+) \boldsymbol{\omega}_+ \right) + \frac{1}{2} \boldsymbol{\omega}^\top \nabla_{\boldsymbol{\omega}}^2 r(\mathbf{x}; \boldsymbol{\omega}_+) \boldsymbol{\omega} + \text{const} \tag{27}$$

where $\text{const}$ absorbs all constant vectors (i.e., not dependent on $\boldsymbol{\omega}$). Hence, taking absolute values for both sides of Eq. (27), we obtain

$$\left| g(\mathbf{x}; \boldsymbol{\omega}) \right| = \left| \boldsymbol{\omega}^\top \left( \nabla_{\boldsymbol{\omega}} r(\mathbf{x}; \boldsymbol{\omega}_+) - \nabla_{\boldsymbol{\omega}}^2 r(\mathbf{x}; \boldsymbol{\omega}_+) \boldsymbol{\omega}_+ \right) + \frac{1}{2} \boldsymbol{\omega}^\top \nabla_{\boldsymbol{\omega}}^2 r(\mathbf{x}; \boldsymbol{\omega}_+) \boldsymbol{\omega} + \text{const} \right| \tag{28}$$

$$\leq \|\boldsymbol{\omega}\| \cdot \|\nabla_{\boldsymbol{\omega}} r(\mathbf{x}; \boldsymbol{\omega}_+) - \nabla_{\boldsymbol{\omega}}^2 r(\mathbf{x}; \boldsymbol{\omega}_+) \boldsymbol{\omega}_+\| + \frac{1}{2} \left| \boldsymbol{\omega}^\top \nabla_{\boldsymbol{\omega}}^2 r(\mathbf{x}; \boldsymbol{\omega}_+) \boldsymbol{\omega} \right| + \|\text{const}\| \tag{29}$$

$$\leq \tau \cdot \|\text{const}\| + \left( \frac{1}{2} \lambda_{\max} \left( \nabla_{\boldsymbol{\omega}}^2 r(\mathbf{x}; \boldsymbol{\omega}_+) \right) \right) \|\boldsymbol{\omega}\|^2 + \|\text{const}\| \tag{30}$$

$$\leq \tau \cdot \|\text{const}\| + \left( \frac{1}{2} \lambda_{\max} \left( \nabla_{\boldsymbol{\omega}}^2 r(\mathbf{x}; \boldsymbol{\omega}_+) \right) \right) \cdot \tau^2 + \|\text{const}\| = \text{const} \tag{31}$$

where Eq. (30) follows from the fact that $\nabla_{\boldsymbol{\omega}}^2 r(\mathbf{x}; \boldsymbol{\omega}_+)$ is positive definite by construction. Eq. (31) follows because $(1/2)\lambda_{\max}(\nabla_{\boldsymbol{\omega}}^2 r(\mathbf{x}; \boldsymbol{\omega}_+))$ is a fixed value that does not change when $\boldsymbol{\omega}$ changes. Thus, $g(\mathbf{x}; \boldsymbol{\omega})$ is bounded as expected. $\square$

# B  Upper-Bound $\mathbb{E}_{\boldsymbol{\delta} \in \mathbb{N}(0, \sigma^2 \mathbf{I})}[F_{\mathcal{X}}(\boldsymbol{\omega} + \boldsymbol{\delta})]$ with $\mathbb{E}_{\boldsymbol{\delta} \in \mathbb{N}(0, \sigma^2 \mathbf{I})}[F_{\mathcal{D}}(\boldsymbol{\omega} + \boldsymbol{\delta})]$

**Lemma 3.** *Let us define*

$$F_{\mathcal{X}}(\boldsymbol{\omega} + \boldsymbol{\delta}) \triangleq \left| \mathbb{E}_{\mathbf{x} \in \mathcal{X}}[g(\mathbf{x}; \boldsymbol{\omega} + \boldsymbol{\delta})] - \mathbb{E}_{\mathbf{x} \in \mathcal{X}}[g(\mathbf{x}; \boldsymbol{\omega})] \right|, \tag{32}$$

$$F_{\mathcal{D}}(\boldsymbol{\omega} + \boldsymbol{\delta}) \triangleq \left| \mathbb{E}_{\mathbf{x} \in \mathcal{D}}[g(\mathbf{x}; \boldsymbol{\omega} + \boldsymbol{\delta})] - \mathbb{E}_{\mathbf{x} \in \mathcal{D}}[g(\mathbf{x}; \boldsymbol{\omega})] \right|. \tag{33}$$

*The following holds simultaneously for all $\boldsymbol{\omega}$, $\sigma^2 > 0$, $\alpha \in (0, 1)$ and $m = \dim(\boldsymbol{\omega})$,*

$$\mathbb{E}_{\boldsymbol{\delta} \sim \mathcal{N}(0, \sigma^2 \mathbf{I})}\left[F_{\mathcal{X}}(\boldsymbol{\omega} + \boldsymbol{\delta})\right] \leq \mathbb{E}_{\boldsymbol{\delta} \sim \mathcal{N}(0, \sigma^2 \mathbf{I})}\left[F_{\mathcal{D}}(\boldsymbol{\omega} + \boldsymbol{\delta})\right] \tag{34}$$

$$+ \sqrt{\frac{\frac{1}{4} m \log \left(1 + \frac{\|\boldsymbol{\omega}\|_2^2}{m \sigma^2}\right) + \frac{1}{4} + \frac{1}{2} \log \frac{n}{\alpha} + \log(8n + 4m)}{n - 1}}$$

*with probability at least $1 - \alpha$ over the (random) choice of the offline dataset $\mathcal{D}$.*

**Proof.** We can view the random perturbation $\boldsymbol{\delta}$ as a random hypothesis sampled from a hypothesis distribution (such as the zero-mean Gaussian $\mathbb{N}(0, \sigma^2 \mathbf{I})$) and $F_{\mathcal{X}}(\boldsymbol{\omega} + \boldsymbol{\delta})$ as its incurred loss. In this view, for any hypothesis distribution $\mathcal{Q}$, $\mathbb{E}_{\boldsymbol{\delta} \sim \mathcal{Q}}[F_{\mathcal{X}}(\boldsymbol{\omega} + \boldsymbol{\delta})]$ and $\mathbb{E}_{\boldsymbol{\delta} \sim \mathcal{Q}}[F_{\mathcal{D}}(\boldsymbol{\omega} + \boldsymbol{\delta})]$ denote the corresponding generalized and empirical Gibbs losses whose relationship is characterized via the following classical PAC-Bayesian [18] bound,

$$\mathbb{E}_{\boldsymbol{\delta} \sim \mathcal{Q}}\left[F_{\mathcal{X}}(\boldsymbol{\omega} + \boldsymbol{\delta})\right] \leq \mathbb{E}_{\boldsymbol{\delta} \sim \mathcal{Q}}\left[F_{\mathcal{D}}(\boldsymbol{\omega} + \boldsymbol{\delta})\right] + \sqrt{\frac{\text{KL}(\mathcal{Q}||\mathcal{P}) + \log(\frac{n}{\alpha})}{2(n - 1)}}, \tag{35}$$

which will hold simultaneously for all choices of $\boldsymbol{\omega}$, $\mathcal{P}$ and $\mathcal{Q}$. Historically, in probabilistic learning context, $\mathcal{P}$ is often referred to as prior or reference distribution which, for example, encodes domain-specific information about certain properties of the solution distribution. Then, $\mathcal{Q}$ is the posterior, which can be selected to tighten the bound (hence, reducing the generalized loss).

In our specific context, we will simply use the choices of $\mathcal{P}$ and $\mathcal{Q}$ as technical vehicles to tighten the gap between $\mathbb{E}[F_{\mathcal{X}}(\boldsymbol{\omega} + \boldsymbol{\delta})]$ and $\mathbb{E}[F_{\mathcal{D}}(\boldsymbol{\omega} + \boldsymbol{\delta})]$ which will contribute to the gap between our generalized and empirical surrogate sharpness.

To derive the optimal choices $\mathcal{P}$ and $\mathcal{Q}$, we will leverage part of the proof from [14], which starts with the following fact:

$$\sqrt{\frac{m \log \left(1 + \frac{\|\boldsymbol{\omega}\|_2^2}{m \sigma^2}\right)}{4n}} \leq \sqrt{\frac{\frac{1}{4} m \log \left(1 + \frac{\|\boldsymbol{\omega}\|_2^2}{m \sigma^2}\right) + \frac{1}{4} + \frac{1}{2} \log \frac{n}{\alpha} + \log(8n + 4m)}{n - 1}}. \tag{36}$$

This means if $\|\boldsymbol{\omega}\| > \rho^2(\exp(4n/m) - 1)$, then

$$\sqrt{\frac{m\log\left(1 + \frac{\|\boldsymbol{\omega}\|_2^2)}{m\sigma^2}\right)}{4n}} \quad > \quad 1 \,, \tag{37}$$

which means the bound gap in Eq. (34) is larger than 1 and under Assumption 1 that the surrogate's output – hence, $F_{\mathcal{X}}(\boldsymbol{\omega})$'s – is bounded in $[0, 1]$, Eq. (34) holds trivially.

Hence, to choose $\mathcal{P}$ and $\mathcal{Q}$, we can make the assumption that $\|\boldsymbol{\omega}\| \leq \rho^2(\exp(4n/m)-1)$. Now, let us choose both prior and posterior distribution to be Gaussians, $\mathcal{P} = \mathbb{N}(\boldsymbol{\mu}_{\mathcal{P}}, \sigma_{\mathcal{P}}^2 \mathbf{I})$ and $\mathcal{Q} = \mathbb{N}(\boldsymbol{\mu}_{\mathcal{Q}}, \sigma_{\mathcal{Q}}^2 \mathbf{I})$. Their KL divergence can then be computed in closed-form:

$$\mathrm{KL}(\mathcal{Q}||\mathcal{P}) \quad = \quad \frac{1}{2}\left[\frac{m\sigma_{\mathcal{Q}}^2 + \|\boldsymbol{\mu}_{\mathcal{P}} - \boldsymbol{\mu}_{\mathcal{Q}}\|_2^2}{\sigma_{\mathcal{P}}^2} \; - \; m \; + \; m\log\left(\frac{\sigma_{\mathcal{P}}}{\sigma_{\mathcal{Q}}}\right)^2\right]. \tag{38}$$

Naively, we can minimize the above KL divergence via setting the derivative of KL with respect to $\sigma_{\mathcal{P}}$ to zero, and solving for the optimal $\sigma_{\mathcal{P}}$, which gives $\sigma_{\mathcal{P}}^2 = \sigma_{\mathcal{Q}}^2 + m^{-1}\|\boldsymbol{\mu}_{\mathcal{P}} - \boldsymbol{\mu}_{\mathcal{Q}}\|_2^2$.

However, for the PAC-Bayes bound to hold, $\sigma_{\mathcal{P}}$ must be chosen independent of the rest, so the above approach does not work. Instead, we can define in advance a prior set of $\sigma_{\mathcal{P}}$ and then choose the best one in this set, following the proof in [24]. We can choose this set as follow:

Given fixed $a, b > 0$, let $S = \{c \cdot \exp((1-i)/m)|i \in \mathbb{N}\}$ be predefined set for $\sigma_{\mathcal{P}}^2$. If the above bound holds for $\sigma_{\mathcal{P}}^2 = c \times \exp((1 - i)/m)$ with probability $1 - \alpha_i$ with $\alpha_i = 6\alpha\pi^{-2}i^{-2}$ for any $i \in \mathbb{N}$, then all above bounds hold simultaneously with probability at least $1 - \sum_{i=1}^{\infty} 6\alpha\pi^{-2}i^{-2} = 1 - \alpha$.

Now, let $\sigma_{\mathcal{Q}} = \sigma, \boldsymbol{\mu}_{\mathcal{Q}} = \boldsymbol{\omega}, \boldsymbol{\mu}_{\mathcal{P}} = 0$, we have

$$\sigma_{\mathcal{Q}}^2 \; + \; \|\boldsymbol{\mu}_{\mathcal{P}} - \boldsymbol{\mu}_{\mathcal{Q}}\|_2^2/m \quad = \quad \sigma^2 \; + \; \|\boldsymbol{\omega}\|_2^2/m \quad \leq \quad \sigma^2(1 + \exp(4n/m)) \tag{39}$$

Choosing $i = \lfloor 1 - m\log((\sigma^2 + \|\boldsymbol{\omega}\|_2^2/m)/c)\rfloor, c = \sigma^2(1 + \exp(4n/m))$ and $\sigma_{\mathcal{P}}^2 = c \cdot \exp((1-i)/m)$,

$$-m\log((\sigma^2 + \|\boldsymbol{\omega}\|_2^2/m)/c) \leq \quad i \quad \leq 1 - m\log((\sigma^2 + \|\boldsymbol{\omega}\|_2^2/m)/c) \tag{40}$$

$$(\sigma^2 + \|\boldsymbol{\omega}\|_2^2/m)/c \leq \quad \exp((1-i)/m) \quad \leq \exp(1/m) \times (\sigma^2 + \|\boldsymbol{\omega}\|_2^2/m)/c \tag{41}$$

$$\sigma^2 + \|\boldsymbol{\omega}\|_2^2/m \leq \quad c \times \exp((1-i)/m) \quad \leq \exp(1/m) \times (\sigma^2 + \|\boldsymbol{\omega}\|_2^2/m) \tag{42}$$

$$\sigma^2 + \|\boldsymbol{\omega}\|_2^2/m \leq \quad \sigma_{\mathcal{P}}^2 \quad \leq \exp(1/m) \times (\sigma^2 + \|\boldsymbol{\omega}\|_2^2/m) \tag{43}$$

Plugging Eq. (43) and Eq. (39) into Eq. (38),

$$\mathrm{KL}(\mathcal{Q}||\mathcal{P}) \quad = \quad \frac{1}{2}\left[\frac{m\sigma_{\mathcal{Q}}^2 + \|\boldsymbol{\mu}_{\mathcal{P}} - \boldsymbol{\mu}_{\mathcal{Q}}\|_2^2}{\sigma_{\mathcal{P}}^2} - m + m\log\left(\frac{\sigma_{\mathcal{P}}}{\sigma_{\mathcal{Q}}}\right)^2\right] \tag{44}$$

$$\leq \quad \frac{1}{2}\left[\frac{m\left(\sigma^2 + \|\boldsymbol{\omega}\|_2^2/m\right)}{\sigma^2 + \|\boldsymbol{\omega}\|_2^2/m} - m + m\log\left(\frac{\exp\left(\frac{1}{m}\right) \times \left(\sigma^2 + \frac{\|\boldsymbol{\omega}\|_2^2}{m}\right)}{\sigma^2}\right)\right] \tag{45}$$

$$= \quad \frac{1}{2}\left[m\log\left(\frac{\exp\left(\frac{1}{m}\right) \times \left(\sigma^2 + \frac{\|\boldsymbol{\omega}\|_2^2}{m}\right)}{\sigma^2}\right)\right] \tag{46}$$

$$= \quad \frac{1}{2}\left[1 + m\log\left(1 + \frac{\|\boldsymbol{\omega}\|_2^2)}{m\sigma_{\mathcal{Q}}^2}\right)\right] \tag{47}$$

Thus, Eq. (35) holds for each prior (indexed by $i$) in the above set with probability $1 - \alpha_i$. Hence, Eq. (35) holds with all prior choices with probability $1 - \sum_i \alpha_i = 1 - \alpha$. This allows us to pick the prior that leads to the tightest bound gap, which is indexed with $i = \lfloor 1 - m\log((\sigma^2 + \|\boldsymbol{\omega}\|_2^2/m)/c)\rfloor$. For this choice, we have

$$\underset{\boldsymbol{\delta}\sim\mathcal{Q}}{\mathbb{E}}\left[F_{\mathcal{X}}(\boldsymbol{\omega} + \boldsymbol{\delta})\right] \quad \leq \quad \underset{\boldsymbol{\delta}\sim\mathcal{Q}}{\mathbb{E}}\left[F_{\mathcal{D}}(\boldsymbol{\omega} + \boldsymbol{\delta})\right] \quad + \quad \sqrt{\frac{\mathrm{KL}(\mathcal{Q}||\mathcal{P}) + \log(\frac{n}{\alpha_i})}{2(n-1)}} \,, \tag{48}$$

Plugging Eq. (47) into Eq. (48),

$$\mathop{\mathbb{E}}_{\boldsymbol{\delta}\sim\mathcal{N}(0,\sigma^2\mathbf{I})}\left[F_{\mathcal{X}}(\boldsymbol{\omega}+\boldsymbol{\delta})\right] \quad \leq \quad \mathop{\mathbb{E}}_{\boldsymbol{\delta}\sim\mathcal{N}(0,\sigma^2\mathbf{I})}\left[F_{\mathcal{D}}(\boldsymbol{\omega}+\boldsymbol{\delta})\right] \tag{49}$$
$$+ \quad \sqrt{\frac{\frac{1}{4}m\log\left(1+\frac{\|\boldsymbol{\omega}\|_2^2}{m\sigma^2}\right)+\frac{1}{4}+\frac{1}{2}\log\frac{n}{\alpha_i}}{n-1}}$$

Finally, to remove the dependence on $i$ in the above bound, note that

$$\log\frac{n}{\alpha_i} \quad = \quad \log\frac{n}{\alpha}+\log\frac{\pi^2 i^2}{6} \tag{50}$$
$$\leq \quad \log\frac{n}{\alpha}+\log\frac{\pi^2(1-m\log((\sigma^2+\|\boldsymbol{\omega}\|_2^2/m)/c))^2}{6} \tag{51}$$
$$\leq \quad \log\frac{n}{\alpha}+\log\frac{\pi^2 m^2\log^2(c/(\sigma^2+\|\boldsymbol{\omega}\|_2^2/m))}{3} \tag{52}$$
$$\leq \quad \log\frac{n}{\alpha}+\log\frac{\pi^2 m^2\log^2(c/\sigma^2)}{3} \tag{53}$$
$$\leq \quad \log\frac{n}{\alpha}+\log\frac{\pi^2 m^2\log^2(1+\exp(4n/m))}{3} \tag{54}$$
$$\leq \quad \log\frac{n}{\alpha}+\log\frac{\pi^2 m^2(2+4n/m)^2}{3} \tag{55}$$
$$\leq \quad \log\frac{n}{\alpha}+2\log\frac{\pi(2m+4n)}{\sqrt{3}} \tag{56}$$
$$\leq \quad \log\frac{n}{\alpha}+2\log(8n+4m) \tag{57}$$

Plugging Eq. (57) into Eq. (49),

$$\mathop{\mathbb{E}}_{\boldsymbol{\delta}\sim\mathcal{N}(0,\sigma^2\mathbf{I})}\left[F_{\mathcal{X}}(\boldsymbol{\omega}+\boldsymbol{\delta})\right] \quad \leq \quad \mathop{\mathbb{E}}_{\boldsymbol{\delta}\sim\mathcal{N}(0,\sigma^2\mathbf{I})}\left[F_{\mathcal{D}}(\boldsymbol{\omega}+\boldsymbol{\delta})\right] \tag{58}$$
$$+ \quad \sqrt{\frac{\frac{1}{4}m\log\left(1+\frac{\|\boldsymbol{\omega}\|_2^2}{m\sigma^2}\right)+\frac{1}{4}+\frac{1}{2}\log\frac{n}{\alpha}+\log(8n+4m)}{n-1}}$$

which completes our proof. $\square$

## C   Upper-Bound $\mathbb{E}_{\boldsymbol{\delta}\in\mathbb{N}(0,\sigma^2\mathbf{I})}[F_{\mathcal{X}}(\boldsymbol{\omega}+\boldsymbol{\delta})]$ with $\mathcal{R}_{\mathcal{D}}(\boldsymbol{\omega})$

**Lemma 4.** *Following the same setup in Lemma 3, the following holds simultaneously for all $\boldsymbol{\omega}$, $\sigma^2>0$, $\alpha\in(0,1)$ and $m=\dim(\boldsymbol{\omega})$,*

$$\mathop{\mathbb{E}}_{\boldsymbol{\delta}\sim\mathcal{N}(0,\sigma^2\mathbf{I})}\left[F_{\mathcal{X}}(\boldsymbol{\omega}+\boldsymbol{\delta})\right] \quad \leq \quad \max_{\|\boldsymbol{\delta}\|\leq\rho}\left[F_{\mathcal{D}}(\boldsymbol{\omega}+\boldsymbol{\delta})\right] \tag{59}$$
$$+ \quad \sqrt{\frac{m\log\left(1+\frac{\|\boldsymbol{\omega}\|_2^2}{m\sigma^2}\right)+2\log\frac{n}{\alpha}+4\log(8n+4m)}{n-1}}$$

*with probability at least $1-\alpha$ over the (random) choice of the offline dataset $\mathcal{D}$.*

**Proof.** To derive this result, we need to upper-bound $\mathbb{E}[F_{\mathcal{X}}(\boldsymbol{\omega})]$ with $F_{\mathcal{D}}(\boldsymbol{\omega})$. First, we note that $\boldsymbol{\delta}\sim\mathcal{N}(0,\sigma^2\mathbf{I})$ so $\|\boldsymbol{\delta}\|_2^2$ follows a chi-square distribution. From Lemma 1 in [25],

$$P\left(\|\boldsymbol{\delta}\|_2^2-m\sigma^2 \geq 2\sigma^2\sqrt{mt}+2t\sigma^2\right) \quad \leq \quad \exp(-t) \quad \forall t>0\,. \tag{60}$$

Therefore, with probability $1-1/\sqrt{n}$ we have:

$$\|\boldsymbol{\delta}\|_2^2 \quad \leq \quad \sigma^2\left(2\log\left(\sqrt{n}\right)+m+2\sqrt{m\log\left(\sqrt{n}\right)}\right) \quad \leq \quad \sigma^2 m\left(1+\sqrt{\frac{\log(n)}{m}}\right)^2 \quad \leq \quad \rho^2\,. \tag{61}$$

This means with probability $1 - 1/\sqrt{n}$, we have $\mathbb{E}[F_{\mathcal{D}}(\boldsymbol{\omega} + \boldsymbol{\delta})] \leq \max_{\|\boldsymbol{\delta}\| \leq \rho} F_{\mathcal{D}}(\boldsymbol{\omega} + \boldsymbol{\delta})$. Otherwise, with the remaining $1/\sqrt{n}$ chance, $\mathbb{E}[F_{\mathcal{D}}(\boldsymbol{\omega} + \boldsymbol{\delta})] \leq 1$ under Assumption 1. Putting this together,

$$\underset{\boldsymbol{\delta} \sim \mathcal{N}(0, \sigma^2 \mathbf{I})}{\mathbb{E}}[F_{\mathcal{D}}(\boldsymbol{\omega} + \boldsymbol{\delta})] \quad \leq \quad \left(1 - \frac{1}{\sqrt{n}}\right) \cdot \max_{\|\boldsymbol{\delta}\| \leq \rho}[F_{\mathcal{D}}(\boldsymbol{\omega} + \boldsymbol{\delta})] \ + \ \frac{1}{\sqrt{n}} \ . \tag{62}$$

Finally, plugging Eq. (62) into Eq. (34),

$$
\begin{aligned}
\underset{\boldsymbol{\delta} \sim \mathcal{N}(0, \sigma^2 \mathbf{I})}{\mathbb{E}}\Big[F_{\mathcal{X}}(\boldsymbol{\omega} + \boldsymbol{\delta})\Big] \quad &\leq \quad \left(1 - \frac{1}{\sqrt{n}}\right) \max_{\|\boldsymbol{\delta}\|_2 \leq \rho} F_{\mathcal{D}}(\boldsymbol{\omega} + \boldsymbol{\delta}) \ + \ \frac{1}{\sqrt{n}} \\
&\quad + \ \sqrt{\frac{\frac{1}{4}m \log\left(1 + \frac{\|\boldsymbol{\omega}\|_2^2)}{m\sigma^2}\left(1 + \sqrt{\frac{\log(n)}{m}}\right)^2\right) + \frac{1}{2}\log\frac{n}{\alpha} + \log(8n + 4m)}{n - 1}} \\
&\leq \quad \max_{\|\boldsymbol{\delta}\|_2 \leq \rho} F_{\mathcal{D}}(\boldsymbol{\omega} + \boldsymbol{\delta}) \\
&\quad + \ \sqrt{\frac{m \log\left(1 + \frac{\|\boldsymbol{\omega}\|_2^2)}{m\sigma^2}\left(1 + \sqrt{\frac{\log(n)}{m}}\right)^2\right) + 2\log\frac{n}{\alpha} + 4\log(8n + 4m)}{n - 1}}
\end{aligned}
$$

which completes our proof. $\square$

# D  Upper-Bound $\mathcal{R}_{\mathcal{X}}(\boldsymbol{\omega})$ with a scaled value of $\mathbb{E}_{\boldsymbol{\delta} \in \mathbb{N}(0, \sigma^2 \mathbf{I})}[F_{\mathcal{X}}(\boldsymbol{\omega} + \boldsymbol{\delta})]$

**Lemma 5.** *Following the same setup in Lemma 3, the below holds for all $\sigma^2 \in (0, 2/(m\lambda_{\min}))$,*

$$\max_{\|\boldsymbol{\delta}\| \leq \rho} F_{\mathcal{X}}(\boldsymbol{\omega} + \boldsymbol{\delta}) \leq \frac{1}{m} \cdot \frac{1}{\sigma^2} \cdot \frac{1}{\lambda_{\min}} \cdot \left(2G(\boldsymbol{\omega})\rho \ + \ \lambda_{\max}\rho^2\right) \cdot \underset{\boldsymbol{\delta} \sim \mathcal{N}(0, \sigma^2 \mathbf{I})}{\mathbb{E}}\Big[F_{\mathcal{X}}(\boldsymbol{\omega} + \boldsymbol{\delta})\Big] , \tag{63}$$

*with $G(\boldsymbol{\omega})$ and $\lambda_{\max}$ defined previously in Eq. (7), $m = \dim(\boldsymbol{\gamma})$ and $\lambda_{\min}$ defined in Assumption 2.*

**Proof.** Let $h(\boldsymbol{\omega}) \triangleq \mathbb{E}[g(\mathbf{x}; \boldsymbol{\omega})]$ where the expectation is over $\mathbf{x} \in \mathcal{X}$, we have

$$
\begin{aligned}
\max_{\|\boldsymbol{\delta}\| \leq \rho} F_{\mathcal{X}}(\boldsymbol{\omega} + \boldsymbol{\delta}) &= \max_{\|\boldsymbol{\delta}\| \leq \rho} |h(\boldsymbol{\omega} + \boldsymbol{\delta}) - h(\boldsymbol{\omega})| \tag{64} \\
&= \max_{\|\boldsymbol{\delta}\| \leq \rho} \left|\nabla_{\boldsymbol{\omega}} h(\boldsymbol{\omega})^\top \boldsymbol{\delta} \ + \ \frac{1}{2}\boldsymbol{\delta}^\top \nabla_{\boldsymbol{\omega}}^2 h(\hat{\boldsymbol{\omega}})\boldsymbol{\delta}\right| \tag{65} \\
&\leq \max_{\|\boldsymbol{\delta}\| \leq \rho} \left|\nabla_{\boldsymbol{\omega}} h(\boldsymbol{\omega})^\top \boldsymbol{\delta}\right| + \frac{1}{2}\left|\boldsymbol{\delta}^\top \nabla_{\boldsymbol{\omega}}^2 h(\hat{\boldsymbol{\omega}})\boldsymbol{\delta}\right| \tag{66} \\
&\leq \max_{\|\boldsymbol{\delta}\| \leq \rho} \|\nabla_{\boldsymbol{\omega}} h(\boldsymbol{\omega})\| \cdot \|\boldsymbol{\delta}\| \ + \ \frac{1}{2}\|\boldsymbol{\delta}\|^2 \cdot \lambda_{\max} \tag{67} \\
&\leq \|\nabla_{\boldsymbol{\omega}} h(\boldsymbol{\omega})\| \, \rho \ + \ \frac{1}{2}\rho^2 \lambda_{\max} \ = \ G(\boldsymbol{\omega})\rho \ + \ \frac{1}{2}\rho^2 \lambda_{\max} \tag{68}
\end{aligned}
$$

with $\hat{\boldsymbol{\omega}} = \boldsymbol{\omega} + c \cdot \boldsymbol{\delta}$ with some constant $c \in [0, 1]$. In the above, Eq. (65) follows from the Taylor's remainder theorem for multivariate function. Eq. (67) follows from the Cauchy-Schwartz inequality (for the first term) and the definition of $\lambda_{\max}$ as the upper-bound on the largest eigenvalue of the parameter Hessian of $h(\boldsymbol{\omega})$. Note that $\lambda_{\max} > \lambda_{\min} > 0$ under Assumption 2 which stipulates that the smallest eigenvalue of the parameter Hessian of $h(\boldsymbol{\omega})$ is always positive.

On the other hand, we also:

$$\mathbb{E}_{\boldsymbol{\delta}\sim\mathbb{N}(0,\sigma^2\mathbf{I})}\Big[F_{\mathcal{X}}(\boldsymbol{\omega}+\boldsymbol{\delta})\Big] = \mathbb{E}_{\boldsymbol{\delta}\sim\mathbb{N}(0,\sigma^2\mathbf{I})}\Big|h(\boldsymbol{\omega}+\boldsymbol{\delta}) - h(\boldsymbol{\omega})\Big| \tag{69}$$

$$\geq \left|\mathbb{E}_{\boldsymbol{\delta}\sim\mathbb{N}(0,\sigma^2\mathbf{I})}\left[\nabla_{\boldsymbol{\omega}}h(\boldsymbol{\omega})^{\top}\boldsymbol{\delta} + \frac{1}{2}\boldsymbol{\delta}^{\top}\nabla_{\boldsymbol{\omega}}^2 h(\hat{\boldsymbol{\omega}})\boldsymbol{\delta}\right]\right| \tag{70}$$

$$= \frac{1}{2}\left|\mathbb{E}_{\boldsymbol{\delta}\sim\mathbb{N}(0,\sigma^2\mathbf{I})}\left[\boldsymbol{\delta}^{\top}\nabla_{\boldsymbol{\omega}}^2 h(\hat{\boldsymbol{\omega}})\boldsymbol{\delta}\right]\right| \tag{71}$$

$$\geq \left(\frac{1}{2}\lambda_{\min}\right)\mathbb{E}_{\boldsymbol{\delta}\sim\mathbb{N}(0,\sigma^2\mathbf{I})}\big[\|\boldsymbol{\delta}\|^2\big] \tag{72}$$

$$= \left(\frac{1}{2}\lambda_{\min}\right)\mathrm{Tr}\left[\sigma^2\mathbf{I}\right] = \left(\frac{1}{2}\lambda_{\min}\right)\cdot m\cdot\sigma^2 \tag{73}$$

where $m = \dim(\boldsymbol{\omega})$. In the above, Eq. (70) follows from the Taylor's remainder theorem (similar to Eq. (65)) and Eq. (72) follows from the definition of $\lambda_{\min}$ in Assumption 2 as the lower-bound on the smallest eigenvalue of the parameter Hessian. Since $\lambda_{\min} > 0$, the quadratic term inside the expectation is always positive which explains why we can remove the absolute operator. Eq. (73) follows from standard moment calculation of Gaussian random vector.

Finally, we note that:

$$G(\boldsymbol{\omega})\rho + \frac{1}{2}\rho^2\lambda_{\max} = \left[\frac{1}{m}\cdot\frac{1}{\sigma^2}\cdot\frac{1}{\lambda_{\min}}\cdot\left(2G(\boldsymbol{\omega})\rho + \lambda_{\max}\rho^2\right)\right]\left(\frac{1}{2}\lambda_{\min}\right)\cdot m\cdot\sigma^2 \tag{74}$$

$$\leq \left[\frac{1}{m}\cdot\frac{1}{\sigma^2}\cdot\frac{1}{\lambda_{\min}}\cdot\left(2G(\boldsymbol{\omega})\rho + \lambda_{\max}\rho^2\right)\right]\cdot\mathbb{E}_{\boldsymbol{\delta}\sim\mathbb{N}(0,\sigma^2\mathbf{I})}\Big[F_{\mathcal{X}}(\boldsymbol{\omega}+\boldsymbol{\delta})\Big] \tag{75}$$

which means, intuitively,

$$\xi = \left[\frac{1}{m}\cdot\frac{1}{\sigma^2}\cdot\frac{1}{\lambda_{\min}}\cdot\left(2G(\boldsymbol{\omega})\rho + \lambda_{\max}\rho^2\right)\right] \tag{76}$$

is the smallest scaling factor for the lower-bound of $\mathbb{E}_{\boldsymbol{\delta}\sim\mathbb{N}(0,\sigma^2\mathbf{I})}[F_{\mathcal{X}}(\boldsymbol{\omega}+\boldsymbol{\delta})]$ so that it matches the upper-bound of $\mathcal{R}_{\mathcal{X}}(\boldsymbol{\omega}) = \max_{\|\boldsymbol{\delta}\|\leq\rho} F_{\mathcal{X}}(\boldsymbol{\omega}+\boldsymbol{\delta})$, as outlined in the 3rd step of our proving plan for Theorem 2. As such, plugging Eq. (68) into the left-hand side of Eq. (75) completes our proof. $\square$

**Remark.** In the above, Eq. (73) suggests that $\mathbb{E}_{\boldsymbol{\delta}}[F_{\mathcal{X}}(\boldsymbol{\omega}+\boldsymbol{\delta})] \geq (1/2)\lambda_{\min}\cdot m\cdot\sigma^2$. But, under Assumption 1, we know that the surrogate's output – and hence $F_{\mathcal{X}}(\boldsymbol{\omega}+\boldsymbol{\delta})$'s – is bounded within $[0,1]$, which means $1 \geq (1/2)\lambda_{\min}\cdot m\cdot\sigma^2$ or equivalently, $\sigma^2 \leq 2/(m\lambda_{\min})$ as stated in Lemma 5's statement. That is, under Assumption 1, Lemma 5 is only correct for $\sigma^2 \in (0, 2/(m\lambda_{\min}))$.

## E    Proof of Theorem 2

We are now ready to prove our main result. First, we re-state the result of Lemma 4 below,

$$\mathbb{E}_{\boldsymbol{\delta}\sim\mathcal{N}(0,\sigma^2\mathbf{I})}\big[F_{\mathcal{X}}(\boldsymbol{\omega}+\boldsymbol{\delta})\big] \leq \max_{\|\boldsymbol{\delta}\|\leq\rho}\big[F_{\mathcal{D}}(\boldsymbol{\omega}+\boldsymbol{\delta})\big]$$

$$+ \sqrt{\frac{m\log\left(1 + \frac{\|\boldsymbol{\omega}\|_2^2)}{m\sigma^2}\right) + 2\log\frac{n}{\alpha} + 4\log(8n+4m)}{n-1}} \tag{77}$$

$$= R_{\mathcal{D}}(\boldsymbol{\omega})$$

$$+ \sqrt{\frac{m\log\left(1 + \frac{\|\boldsymbol{\omega}\|_2^2)}{m\sigma^2}\right) + 2\log\frac{n}{\alpha} + 4\log(8n+4m)}{n-1}} \tag{78}$$

where the last step follows from the definition of $\mathcal{R}_{\mathcal{D}}(\boldsymbol{\omega})$. Finally, plugging Eq. (78) into Eq. (63) and replacing $\max_{\|\boldsymbol{\delta}\|\leq\rho} F_{\mathcal{X}}(\boldsymbol{\omega}+\boldsymbol{\delta})$ with $\mathcal{R}_{\mathcal{X}}(\boldsymbol{\omega})$ (following its definition) completes our proof. $\square$

# F   Effective Approximation of $\nabla_{\boldsymbol{\omega}}\|\nabla_{\boldsymbol{\omega}}h(\boldsymbol{\omega})\|$

Explicitly, the gradient of the augmented loss in Eq. (12) is given below,

$$\nabla_{\boldsymbol{\omega}}\mathcal{L}(\boldsymbol{\omega}, \lambda) = \nabla_{\boldsymbol{\omega}}\mathcal{L}_{\mathcal{D}}(\boldsymbol{\omega}) + \lambda \cdot \rho \cdot \nabla_{\boldsymbol{\omega}}\|\nabla_{\boldsymbol{\omega}}h(\boldsymbol{\omega})\|. \tag{79}$$

where the computation bottleneck lies with the second gradient term which can be further expressed below using the chain rule,

$$\nabla_{\boldsymbol{\omega}}\|\nabla_{\boldsymbol{\omega}}h(\boldsymbol{\omega})\| = \nabla_{\boldsymbol{\omega}}^2 h(\boldsymbol{\omega}) \cdot \left(\nabla_{\boldsymbol{\omega}}h(\boldsymbol{\omega}) \,/\, \|\nabla_{\boldsymbol{\omega}}h(\boldsymbol{\omega})\|\right), \tag{80}$$

It is evident that $\nabla_{\boldsymbol{\omega}}^2 h(\boldsymbol{\omega})$ in Eq. (80) represents a Hessian matrix. Calculating the Hessian matrix for a deep neural network is impractical due to the extensive dimensions of the model's weights. Nevertheless, since Eq. (80) involves the multiplication of a Hessian matrix with a vector, specific techniques like Hessian-vector products can be employed to approximate this product. In particular, let the Hessian matrix $\mathbf{M} = \nabla_{\boldsymbol{\omega}}^2 h(\boldsymbol{\omega})$, we have Taylor expansion for function $\nabla_{\boldsymbol{\omega}}h(\boldsymbol{\omega})$ as follows:

$$\nabla_{\boldsymbol{\omega}}h(\boldsymbol{\omega} + \Delta\boldsymbol{\omega}) = \nabla_{\boldsymbol{\omega}}h(\boldsymbol{\omega}) + \mathbf{M}\Delta\boldsymbol{\omega} + \mathcal{O}(\|\Delta\boldsymbol{\omega}\|^2). \tag{81}$$

This approximation becomes precise as the value of $\Delta\boldsymbol{\omega}$ approaches 0. Following [26] and [27], we choose $\Delta\boldsymbol{\omega} = r\mathbf{v}$ where $r$ is a small scalar and $\mathbf{v}$ is a vector, that transforms Eq. (81) into

$$\mathbf{M}\mathbf{v} = \frac{1}{r}\left(\nabla_{\boldsymbol{\omega}}h(\boldsymbol{\omega} + \Delta\boldsymbol{\omega}) - \nabla_{\boldsymbol{\omega}}h(\boldsymbol{\omega})\right) + \mathcal{O}(r). \tag{82}$$

Then, choosing $\mathbf{v} = \frac{\nabla_{\boldsymbol{\omega}}h(\boldsymbol{\omega})}{\|\nabla_{\boldsymbol{\omega}}h(\boldsymbol{\omega})\|}$ leads to

$$\nabla_{\boldsymbol{\omega}}\|\nabla_{\boldsymbol{\omega}}h(\boldsymbol{\omega})\| = \mathbf{M}\frac{\nabla_{\boldsymbol{\omega}}h(\boldsymbol{\omega})}{\|\nabla_{\boldsymbol{\omega}}h(\boldsymbol{\omega})\|} \simeq \frac{1}{r}\left(\nabla_{\boldsymbol{\omega}}h\left(\boldsymbol{\omega} + r\frac{\nabla_{\boldsymbol{\omega}}h(\boldsymbol{\omega})}{\|\nabla_{\boldsymbol{\omega}}h(\boldsymbol{\omega})\|}\right) - \nabla_{\boldsymbol{\omega}}h(\boldsymbol{\omega})\right). \tag{83}$$

It is noticed that, as pointed out by [27], an inappropriate choice of $r$ can make Eq. (83) vulnerable to numeric and roundoff issues. The constant $r$ must be sufficiently small so that the $\mathcal{O}(r)$ term becomes negligible. However, when $r$ is too small, precision is compromised because the subtraction of the original gradient from the perturbed one, i.e., $\nabla_{\boldsymbol{\omega}}h(\boldsymbol{\omega} + r\nabla_{\boldsymbol{\omega}}h(\boldsymbol{\omega})/\|\nabla_{\boldsymbol{\omega}}h(\boldsymbol{\omega})\|) - \nabla_{\boldsymbol{\omega}}h(\boldsymbol{\omega})$, will obtain a small difference between them. Based on Eq. (83), Eq. (79) would be

$$\nabla_{\boldsymbol{\omega}}\mathcal{L}(\boldsymbol{\omega}) = \nabla_{\boldsymbol{\omega}}\mathcal{L}_{\mathcal{D}}(\boldsymbol{\omega}) + \frac{\lambda\rho}{r}\left(\nabla_{\boldsymbol{\omega}}h\left(\boldsymbol{\omega} + r\frac{\nabla_{\boldsymbol{\omega}}h(\boldsymbol{\omega})}{\|\nabla_{\boldsymbol{\omega}}h(\boldsymbol{\omega})\|}\right) - \nabla_{\boldsymbol{\omega}}h(\boldsymbol{\omega})\right). \tag{84}$$

In practical applications, we typically employ an additional approximation to compute the second term in Eq. (84), thereby avoiding the need for Hessian computation induced by the chain rule.

$$\nabla_{\boldsymbol{\omega}}h\left(\boldsymbol{\omega} + r\frac{\nabla_{\boldsymbol{\omega}}h(\boldsymbol{\omega})}{\|\nabla_{\boldsymbol{\omega}}h(\boldsymbol{\omega})\|}\right) \simeq \nabla_{\boldsymbol{\omega}}h(\boldsymbol{\omega})\Big|_{\boldsymbol{\omega} = \boldsymbol{\omega} + r\frac{\nabla_{\boldsymbol{\omega}}h(\boldsymbol{\omega})}{\|\nabla_{\boldsymbol{\omega}}h(\boldsymbol{\omega})\|}}. \tag{85}$$

# G   Hyperparameter Tuning and Additional Experimental Results

## G.1   Computation Resource

All our experiments were conducted on a system with the following specifications: Ubuntu 18.04, NVIDIA RTX 3090 GPUs, and CUDA 11.8.

## G.2   IGNITE-2

**IGNITE-2.** To solve the Lagrangian in Eq. (12), we can treat $\lambda$ as a hyper-parameter and optimize for $\boldsymbol{\omega}$ using stochastic gradient descent (SGD).

$$\boldsymbol{\omega}^{t+1} \leftarrow \boldsymbol{\omega}^t - \eta_{\boldsymbol{\omega}} \cdot \left(\nabla_{\boldsymbol{\omega}}\mathcal{L}_{\mathcal{D}}(\boldsymbol{\omega}^t) + \lambda \cdot \rho \cdot \nabla_{\boldsymbol{\omega}}\|\nabla_{\boldsymbol{\omega}}h(\boldsymbol{\omega}^t)\|\right), \tag{86}$$

where $\boldsymbol{\omega}^t$ is the estimation of the surrogate's parameter at the $t^{th}$ iteration, and $\eta_{\boldsymbol{\omega}}$ is step size to update $\boldsymbol{\omega}$. We name this method **IGNITE-2** and its pseudo-code is in Algorithm 2.

**Hyper-parameter Configuration.** Our method **IGNITE-2** introduces three additional hyper-parameters: $\lambda$, $\rho$, and $r$. The penalty coefficient $\lambda$ controls the gradient magnitude of our regularizer, set to 0.01 through a grid search within $\{0.0001, 0.001, 0.01\}$. The hyper-parameters $\rho$ and $r$ are chosen from $\{0.01, 0.05, 0.1, 0.2, 0.5\}$, with $\rho$ set to 0.2 and $r$ set to 0.2 for **IGNITE**. These three hyper-parameters are determined through experiments in Section G.4. These hyper-parameters are consistently applied across all experiments, except for the **ICT** baseline ($\lambda = 1e - 4$) due to implementation differences.

Table 5: The percentage improvement in performance achieved by **IGNITE-2** and **IGNITE** across all tasks and baseline algorithms at the **100th percentile** level is presented. **Gain** signifies the percentage gain over the baseline performance (Base).

| | | Continuous tasks | | | | Discrete task | | | |
| | | Ant Morphology | | D'Kitty Morphology | | TF Bind 8 | | TF Bind 10 | |
| Algorithms | | Performance | Gain | Performance | Gain | Performance | Gain | Performance | Gain |
|---|---|---|---|---|---|---|---|---|---|
| $\mathfrak{D}(\textbf{best})$ | | 0.565 | | 0.884 | | 0.565 | | 0.884 | |
| REINF-ORCE | Base | $0.255 \pm 0.036$ | | $0.546 \pm 0.208$ | | $0.929 \pm 0.043$ | | $0.635 \pm 0.028$ | |
| | IGNITE-2 | $0.260 \pm 0.037$ | +0.5% | $0.611 \pm 0.176$ | +6.5% | $0.954 \pm 0.027$ | +2.5% | $0.646 \pm 0.028$ | +1.1% |
| | IGNITE | $0.282 \pm 0.021$ | +2.7% | $0.642 \pm 0.160$ | +9.6% | $0.944 \pm 0.030$ | +1.5% | $0.670 \pm 0.060$ | +3.5% |
| GA | Base | $0.303 \pm 0.027$ | | $0.881 \pm 0.016$ | | $0.980 \pm 0.016$ | | $0.651 \pm 0.033$ | |
| | IGNITE-2 | $0.312 \pm 0.038$ | +0.9% | $0.885 \pm 0.022$ | +0.4% | $0.985 \pm 0.007$ | +0.5% | $0.663 \pm 0.090$ | +1.2% |
| | IGNITE | $0.320 \pm 0.044$ | +1.7% | $0.886 \pm 0.017$ | +0.5% | $0.985 \pm 0.010$ | +0.5% | $0.653 \pm 0.043$ | +0.2% |
| ENS-MEAN | Base | $0.376 \pm 0.060$ | | $0.888 \pm 0.010$ | | $0.985 \pm 0.009$ | | $0.649 \pm 0.036$ | |
| | IGNITE-2 | $0.437 \pm 0.068$ | +6.1% | $0.890 \pm 0.010$ | +0.2% | $0.988 \pm 0.005$ | +0.3% | $0.665 \pm 0.091$ | +1.6% |
| | IGNITE | $0.435 \pm 0.058$ | +5.9% | $0.896 \pm 0.013$ | +0.8% | $0.987 \pm 0.007$ | +0.2% | $0.662 \pm 0.091$ | +1.3% |
| ENS-MIN | Base | $0.385 \pm 0.067$ | | $0.889 \pm 0.014$ | | $0.980 \pm 0.012$ | | $0.681 \pm 0.095$ | |
| | IGNITE-2 | $0.441 \pm 0.084$ | +5.6% | $0.894 \pm 0.011$ | +0.5% | $0.982 \pm 0.015$ | +0.2% | $0.686 \pm 0.120$ | +0.5% |
| | IGNITE | $0.468 \pm 0.062$ | +8.3% | $0.897 \pm 0.010$ | +0.8% | $0.986 \pm 0.010$ | +0.6% | $0.705 \pm 0.118$ | +2.4% |
| CbAS | Base | $0.854 \pm 0.042$ | | $0.895 \pm 0.012$ | | $0.919 \pm 0.044$ | | $0.635 \pm 0.041$ | |
| | IGNITE-2 | $0.850 \pm 0.036$ | -0.4% | $0.903 \pm 0.014$ | +0.8% | $0.916 \pm 0.043$ | -0.3% | $0.650 \pm 0.054$ | +1.5% |
| | IGNITE | $0.859 \pm 0.039$ | +0.5% | $0.900 \pm 0.015$ | +0.5% | $0.921 \pm 0.042$ | +0.2% | $0.652 \pm 0.055$ | +1.7% |
| MINs | Base | $0.905 \pm 0.023$ | | $0.944 \pm 0.008$ | | $0.892 \pm 0.046$ | | $0.643 \pm 0.062$ | |
| | IGNITE-2 | $0.907 \pm 0.035$ | +0.2% | $0.940 \pm 0.007$ | -0.4% | $0.915 \pm 0.040$ | +2.3% | $0.645 \pm 0.049$ | +0.2% |
| | IGNITE | $0.911 \pm 0.024$ | +0.6% | $0.945 \pm 0.007$ | +0.1% | $0.930 \pm 0.041$ | +3.8% | $0.647 \pm 0.058$ | +0.4% |
| RoMA | Base | $0.569 \pm 0.086$ | | $0.821 \pm 0.019$ | | $0.665 \pm 0.000$ | | $0.550 \pm 0.008$ | |
| | IGNITE-2 | $0.590 \pm 0.063$ | +2.1% | $0.833 \pm 0.028$ | +1.2% | $0.665 \pm 0.000$ | +0.0% | $0.553 \pm 0.000$ | +0.3% |
| | IGNITE | $0.615 \pm 0.085$ | +4.6% | $0.834 \pm 0.012$ | +1.3% | $0.665 \pm 0.000$ | +0.0% | $0.553 \pm 0.000$ | +0.3% |
| COMs | Base | $0.897 \pm 0.031$ | | $0.931 \pm 0.013$ | | $0.955 \pm 0.030$ | | $0.645 \pm 0.038$ | |
| | IGNITE-2 | $0.911 \pm 0.030$ | +1.4% | $0.940 \pm 0.014$ | +0.9% | $0.948 \pm 0.025$ | -0.7% | $0.637 \pm 0.033$ | -0.8% |
| | IGNITE | $0.901 \pm 0.030$ | +0.4% | $0.934 \pm 0.010$ | +0.3% | $0.952 \pm 0.043$ | -0.3% | $0.638 \pm 0.053$ | -0.7% |
| CMA-ES | Base | $1.955 \pm 1.484$ | | $0.724 \pm 0.002$ | | $0.928 \pm 0.040$ | | $0.668 \pm 0.035$ | |
| | IGNITE-2 | $1.970 \pm 1.971$ | +1.5% | $0.725 \pm 0.006$ | +0.1% | $0.938 \pm 0.031$ | +1.0% | $0.670 \pm 0.033$ | +0.2% |
| | IGNITE | $1.957 \pm 1.910$ | +0.2% | $0.724 \pm 0.001$ | +0.0% | $0.927 \pm 0.043$ | -0.1% | $0.673 \pm 0.044$ | +0.5% |
| BO-qEI | Base | $0.812 \pm 0.000$ | | $0.896 \pm 0.000$ | | $0.787 \pm 0.112$ | | $0.628 \pm 0.000$ | |
| | IGNITE-2 | $0.812 \pm 0.000$ | +0.0% | $0.896 \pm 0.000$ | +0.0% | $0.855 \pm 0.107$ | +6.8% | $0.628 \pm 0.000$ | +0.0% |
| | IGNITE | $0.812 \pm 0.000$ | +0.0% | $0.896 \pm 0.000$ | +0.0% | $0.843 \pm 0.109$ | +5.6% | $0.628 \pm 0.000$ | +0.0% |
| ICT | Base | $0.937 \pm 0.023$ | | $0.946 \pm 0.014$ | | $0.892 \pm 0.055$ | | $0.647 \pm 0.025$ | |
| | IGNITE-2 | $0.936 \pm 0.017$ | -0.1% | $0.947 \pm 0.019$ | +0.1% | $0.920 \pm 0.035$ | +2.8% | $0.656 \pm 0.029$ | +0.9% |
| | IGNITE | $0.935 \pm 0.032$ | -0.2% | $0.962 \pm 0.018$ | +1.6% | $0.923 \pm 0.038$ | +3.1% | $0.652 \pm 0.074$ | +0.5% |

---

**Algorithm 2 IGNITE-2**

**Input:** offline data $\mathcal{D} = \{(\mathbf{x}_i, z_i)\}_{i=1}^{n}$; initial surrogate $g(\mathbf{x}; \boldsymbol{\omega}^{(0)})$; no. of iterations $T$; batch size $m$; Lagrange multiplier $\lambda$; perturbation radius $\rho$ and scalar $r$; step sizes $\eta_{\boldsymbol{\omega}}$.

1: Initialize $\boldsymbol{\omega}^{(1)} \leftarrow \boldsymbol{\omega}^{(0)}$
2: **for** $t \leftarrow 1 : T$ **do**
3:     Sample $\mathcal{B} = \{(\mathbf{x}_i, z_i)\}_{i=1}^{m} \sim \mathcal{D}$
4:     Compute $\hat{z}_i = g(\mathbf{x}_i; \boldsymbol{\omega}^{(t)})$ for $i \in [m]$
5:     Compute $g_1 = m^{-1} \sum_{i=1}^{m} \nabla_{\boldsymbol{\omega}} \ell(\hat{z}_i, z_i)$
6:     Compute $g_2 = m^{-1} \sum_{i=1}^{m} \nabla_{\boldsymbol{\omega}} \hat{z}_i$
7:     Compute $\hat{\boldsymbol{\omega}} = \boldsymbol{\omega}^{(t)} + r \cdot g_2 / \|g_2\|$
8:     Compute $g_3 = m^{-1} \sum_{i=1}^{m} \nabla_{\boldsymbol{\omega}} g(\mathbf{x}_i; \hat{\boldsymbol{\omega}})$
9:     Compute $g^{(t)} = g_1 + \lambda \rho r^{-1}(g_3 - g_2)$
10:     Update $\boldsymbol{\omega}^{(t+1)} \leftarrow \boldsymbol{\omega}^{(t)} - \eta_{\boldsymbol{\omega}} g^{(t)}$
11: **end for**
12: **return** learned surrogate $\boldsymbol{\omega}^{(T+1)}$

### G.3 Performance Evaluation at 100-th, 80-th and 50-th Percentile Level of IGNITE and IGNITE-2

In this section, we presented the percentage improvement over baseline performance attained by **IGNITE-2** and **IGNITE** when it is applied to an existing baseline. We have evaluated this at the 100-th, 80-th, and 50-th percentile levels. The results are reported in Table 5, 6, and 7, respectively.

Table 6: The percentage improvement in performance achieved by **IGNITE-2** and **IGNITE** across all tasks and baseline algorithms at the **80th percentile** level is presented. **Gain** signifies the percentage gain over the baseline performance (Base).

| Algorithms | | Continuous tasks | | | | Discrete task | | | |
|---|---|---|---|---|---|---|---|---|---|
| | | Ant Morphology | | D'Kitty Morphology | | TF Bind 8 | | TF Bind 10 | |
| | | Performance | Gain | Performance | Gain | Performance | Gain | Performance | Gain |
| $\mathfrak{D}$(**best**) | | 0.565 | | 0.884 | | 0.565 | | 0.884 | |
| REINF-ORCE | Base | $0.185 \pm 0.035$ | | $0.508 \pm 0.200$ | | $0.613 \pm 0.029$ | | $0.523 \pm 0.008$ | |
| | IGNITE-2 | $0.191 \pm 0.033$ | +0.6% | $0.462 \pm 0.199$ | -4.6% | $0.620 \pm 0.031$ | +0.7% | $0.520 \pm 0.006$ | -0.3% |
| | IGNITE | $0.210 \pm 0.039$ | +2.5% | $0.532 \pm 0.197$ | +2.4% | $0.616 \pm 0.038$ | +0.3% | $0.523 \pm 0.007$ | +0.0% |
| GA | Base | $0.195 \pm 0.011$ | | $0.784 \pm 0.030$ | | $0.826 \pm 0.032$ | | $0.517 \pm 0.006$ | |
| | IGNITE-2 | $0.189 \pm 0.019$ | -0.6% | $0.794 \pm 0.039$ | +1.0% | $0.832 \pm 0.032$ | +0.6% | $0.518 \pm 0.007$ | +0.1% |
| | IGNITE | $0.201 \pm 0.017$ | +0.6% | $0.810 \pm 0.026$ | +2.6% | $0.833 \pm 0.021$ | +0.7% | $0.517 \pm 0.006$ | +0.0% |
| ENS-MEAN | Base | $0.236 \pm 0.013$ | | $0.823 \pm 0.013$ | | $0.852 \pm 0.023$ | | $0.515 \pm 0.006$ | |
| | IGNITE-2 | $0.235 \pm 0.010$ | -0.1% | $0.835 \pm 0.013$ | +1.2% | $0.855 \pm 0.020$ | +0.3% | $0.511 \pm 0.006$ | -0.4% |
| | IGNITE | $0.244 \pm 0.013$ | +0.8% | $0.841 \pm 0.013$ | +1.8% | $0.848 \pm 0.016$ | -0.4% | $0.517 \pm 0.006$ | +0.2% |
| ENS-MIN | Base | $0.229 \pm 0.016$ | | $0.819 \pm 0.017$ | | $0.845 \pm 0.021$ | | $0.512 \pm 0.005$ | |
| | IGNITE-2 | $0.236 \pm 0.011$ | +0.7% | $0.835 \pm 0.012$ | +1.6% | $0.854 \pm 0.026$ | +0.9% | $0.514 \pm 0.004$ | +0.2 % |
| | IGNITE | $0.249 \pm 0.015$ | +2.0% | $0.845 \pm 0.010$ | +2.6% | $0.834 \pm 0.015$ | -1.1% | $0.516 \pm 0.006$ | +0.4% |
| CbAS | Base | $0.581 \pm 0.037$ | | $0.814 \pm 0.011$ | | $0.576 \pm 0.022$ | | $0.510 \pm 0.009$ | |
| | IGNITE-2 | $0.571 \pm 0.028$ | -1.0% | $0.815 \pm 0.009$ | +0.1% | $0.565 \pm 0.025$ | -1.1% | $0.512 \pm 0.009$ | +0.2 % |
| | IGNITE | $0.550 \pm 0.052$ | -3.1% | $0.812 \pm 0.013$ | -0.2% | $0.565 \pm 0.023$ | -1.1% | $0.512 \pm 0.009$ | +0.2% |
| MINs | Base | $0.754 \pm 0.026$ | | $0.907 \pm 0.003$ | | $0.544 \pm 0.029$ | | $0.518 \pm 0.008$ | |
| | IGNITE-2 | $0.750 \pm 0.023$ | -0.4% | $0.909 \pm 0.003$ | +0.2% | $0.543 \pm 0.028$ | -0.1% | $0.518 \pm 0.009$ | +0.0 % |
| | IGNITE | $0.753 \pm 0.024$ | -0.1% | $0.909 \pm 0.004$ | +0.2% | $0.551 \pm 0.030$ | +0.7% | $0.515 \pm 0.009$ | -0.3% |
| RoMA | Base | $0.306 \pm 0.036$ | | $0.736 \pm 0.016$ | | $0.644 \pm 0.037$ | | $0.527 \pm 0.008$ | |
| | IGNITE-2 | $0.305 \pm 0.031$ | -0.1% | $0.746 \pm 0.022$ | +1.0% | $0.646 \pm 0.049$ | +0.2% | $0.526 \pm 0.003$ | -0.1 % |
| | IGNITE | $0.312 \pm 0.032$ | +0.6% | $0.737 \pm 0.019$ | +0.1% | $0.655 \pm 0.014$ | +1.1% | $0.527 \pm 0.002$ | +0.0% |
| COMs | Base | $0.629 \pm 0.028$ | | $0.887 \pm 0.005$ | | $0.747 \pm 0.052$ | | $0.517 \pm 0.009$ | |
| | IGNITE-2 | $0.653 \pm 0.026$ | +2.4% | $0.889 \pm 0.003$ | +0.2% | $0.742 \pm 0.061$ | -0.5% | $0.519 \pm 0.007$ | +0.2 % |
| | IGNITE | $0.629 \pm 0.025$ | +0.0% | $0.891 \pm 0.005$ | +0.4% | $0.751 \pm 0.063$ | +0.4% | $0.517 \pm 0.013$ | +0.0% |
| CMA-ES | Base | $0.013 \pm 0.017$ | | $0.718 \pm 0.001$ | | $0.653 \pm 0.020$ | | $0.546 \pm 0.014$ | |
| | IGNITE-2 | $0.011 \pm 0.018$ | -0.2% | $0.719 \pm 0.001$ | +0.1% | $0.649 \pm 0.022$ | -0.4% | $0.545 \pm 0.010$ | -0.1 % |
| | IGNITE | $0.013 \pm 0.024$ | +0.0% | $0.718 \pm 0.002$ | +0.0% | $0.652 \pm 0.021$ | -0.1% | $0.549 \pm 0.014$ | +0.3% |
| BO-qEI | Base | $0.629 \pm 0.000$ | | $0.884 \pm 0.000$ | | $0.439 \pm 0.000$ | | $0.513 \pm 0.001$ | |
| | IGNITE-2 | $0.629 \pm 0.000$ | +0.0% | $0.884 \pm 0.000$ | +0.0% | $0.439 \pm 0.000$ | +0.0% | $0.513 \pm 0.001$ | +0.0 % |
| | IGNITE | $0.629 \pm 0.000$ | +0.0% | $0.884 \pm 0.000$ | +0.0% | $0.439 \pm 0.000$ | +0.0% | $0.513 \pm 0.000$ | +0.0% |
| ICT | Base | $0.710 \pm 0.022$ | | $0.896 \pm 0.005$ | | $0.687 \pm 0.054$ | | $0.549 \pm 0.016$ | |
| | IGNITE-2 | $0.702 \pm 0.024$ | -0.8% | $0.897 \pm 0.004$ | +0.0% | $0.680 \pm 0.037$ | -0.7% | $0.551 \pm 0.022$ | +0.2% |
| | IGNITE | $0.668 \pm 0.026$ | -4.2% | $0.895 \pm 0.005$ | -0.1% | $0.651 \pm 0.039$ | -3.6% | $0.503 \pm 0.043$ | -4.6% |

### G.3.1 Performance Evaluation at 100-th Percentile Level of IGNITE-2

According to the reported results in Table 5, **IGNITE-2** consistently maintains a high probability of 86.36% (38 out of 44) of not degrading baseline performance. There is a high likelihood (72.27% = 34 out of 44 cases) of improving baseline performance, with an average improvement of approximately 1.39% and a notable peak improvement of 6.8%. Conversely, **IGNITE-2** also exhibits a relatively low probability (13.64% = 6 out of 44 cases) of decreasing performance, with an average degradation of approximately 0.45% and a minor peak degradation of 0.8%.Additionally, there is a minor probability (9.09% = 4 out of 44 cases) of maintaining baseline performance. Furthermore, while **IGNITE-2** demonstrates significant efficiency, the results in Section 5.2 indicate that **IGNITE** even outperforms **IGNITE-2** .

### G.3.2 Performance Evaluation at 80-th Percentile Level of IGNITE and IGNITE-2

As shown in Table 6, **IGNITE-2** consistently maintains a high probability high probability of not degrading baseline performance, with a likelihood of 61.36% (27 out of 44 cases). There is a 47.73% chance (21 out of 44 cases) of improving baseline performance, with an average improvement of approximately 0.6% and a peak improvement of 2.4%. On the other hand, **IGNITE-2** also indicates a low likelihood (38.64% = 17 out of 44 cases) of performance decrease, with an average decline of around 0.68% and a maximum decline of 4.6%. Furthermore, there is a slight chance (13.64% = 6 out of 44 cases) of maintaining the baseline performance.

Table 7: The percentage improvement in performance achieved by **IGNITE-2** and **IGNITE** across all tasks and baseline algorithms at the **50th percentile** level is presented. **Gain** signifies the percentage gain over the baseline performance (Base).

| | | Continuous tasks | | | | Discrete task | | | |
|---|---|---|---|---|---|---|---|---|---|
| | | Ant Morphology | | D'Kitty Morphology | | TF Bind 8 | | TF Bind 10 | |
| Algorithms | | Performance | Gain | Performance | Gain | Performance | Gain | Performance | Gain |
| $\mathfrak{D}(\text{best})$ | | 0.565 | | 0.884 | | 0.565 | | 0.884 | |
| REINF-ORCE | Base | 0.142 ± 0.042 | | 0.431 ± 0.183 | | 0.459 ± 0.020 | | 0.469 ± 0.008 | |
| | IGNITE-2 | 0.157 ± 0.031 | +1.5% | 0.432 ± 0.187 | +0.1% | 0.459 ± 0.020 | +0.0% | 0.470 ± 0.005 | +0.1% |
| | IGNITE | 0.167 ± 0.043 | +2.5% | 0.453 ± 0.188 | +2.2% | 0.450 ± 0.018 | -0.9% | 0.472 ± 0.005 | +0.3% |
| GA | Base | 0.147 ± 0.011 | | 0.600 ± 0.145 | | 0.613 ± 0.039 | | 0.467 ± 0.004 | |
| | IGNITE-2 | 0.142 ± 0.017 | -0.5% | 0.623 ± 0.181 | +2.3% | 0.612 ± 0.023 | +0.0% | 0.467 ± 0.003 | +0.0% |
| | IGNITE | 0.155 ± 0.013 | +0.8% | 0.746 ± 0.036 | +14.6% | 0.611 ± 0.026 | -0.2% | 0.469 ± 0.005 | +0.2% |
| ENS-MEAN | Base | 0.189 ± 0.011 | | 0.752 ± 0.026 | | 0.643 ± 0.027 | | 0.468 ± 0.002 | |
| | IGNITE-2 | 0.186 ± 0.010 | -0.3% | 0.780 ± 0.025 | +2.8% | 0.670 ± 0.031 | +2.7% | 0.466 ± 0.004 | -0.2% |
| | IGNITE | 0.192 ± 0.008 | +0.3% | 0.796 ± 0.022 | +4.4% | 0.644 ± 0.037 | +0.1% | 0.467 ± 0.001 | -0.1% |
| ENS-MIN | Base | 0.183 ± 0.013 | | 0.741 ± 0.022 | | 0.655 ± 0.031 | | 0.467 ± 0.001 | |
| | IGNITE-2 | 0.188 ± 0.010 | +0.5% | 0.777 ± 0.025 | +2.9% | 0.658 ± 0.038 | +0.3% | 0.467 ± 0.002 | +0.0% |
| | IGNITE | 0.197 ± 0.015 | +1.4% | 0.805 ± 0.016 | +6.4% | 0.626 ± 0.035 | -2.9% | 0.467 ± 0.002 | +0.0% |
| CbAS | Base | 0.382 ± 0.028 | | 0.751 ± 0.015 | | 0.433 ± 0.015 | | 0.458 ± 0.006 | |
| | IGNITE-2 | 0.374 ± 0.020 | -0.8% | 0.744 ± 0.018 | -0.7% | 0.430 ± 0.020 | -0.3% | 0.459 ± 0.008 | +0.1% |
| | IGNITE | 0.374 ± 0.026 | -0.8% | 0.743 ± 0.018 | -0.8% | 0.427 ± 0.020 | -0.6% | 0.461 ± 0.007 | +0.3% |
| MINs | Base | 0.637 ± 0.035 | | 0.888 ± 0.005 | | 0.417 ± 0.019 | | 0.465 ± 0.007 | |
| | IGNITE-2 | 0.628 ± 0.025 | -0.9% | 0.889 ± 0.004 | +0.1% | 0.421 ± 0.014 | +0.4% | 0.469 ± 0.006 | +0.4% |
| | IGNITE | 0.628 ± 0.027 | -0.9% | 0.888 ± 0.005 | +0.0% | 0.422 ± 0.019 | +0.5% | 0.466 ± 0.008 | +0.1% |
| RoMA | Base | 0.233 ± 0.020 | | 0.612 ± 0.146 | | 0.484 ± 0.045 | | 0.513 ± 0.006 | |
| | IGNITE-2 | 0.228 ± 0.018 | -0.5% | 0.584 ± 0.147 | -2.8% | 0.494 ± 0.044 | +1.0% | 0.514 ± 0.006 | +0.1% |
| | IGNITE | 0.228 ± 0.014 | -0.5% | 0.652 ± 0.094 | +4.0% | 0.503 ± 0.068 | +1.9% | 0.517 ± 0.003 | +0.4% |
| COMs | Base | 0.487 ± 0.020 | | 0.859 ± 0.007 | | 0.588 ± 0.037 | | 0.473 ± 0.013 | |
| | IGNITE-2 | 0.503 ± 0.025 | +1.6% | 0.862 ± 0.004 | +0.3% | 0.577 ± 0.040 | -1.1% | 0.474 ± 0.006 | +0.1% |
| | IGNITE | 0.482 ± 0.027 | -0.5% | 0.862 ± 0.008 | +0.3% | 0.598 ± 0.042 | +1.0% | 0.471 ± 0.010 | -0.2% |
| CMA-ES | Base | -0.043 ± 0.008 | | 0.677 ± 0.014 | | 0.538 ± 0.013 | | 0.492 ± 0.013 | |
| | IGNITE-2 | -0.045 ± 0.008 | -0.2% | 0.685 ± 0.011 | +0.8% | 0.532 ± 0.020 | -0.6% | 0.493 ± 0.012 | +0.1% |
| | IGNITE | -0.049 ± 0.009 | -0.6% | 0.681 ± 0.014 | +0.4% | 0.542 ± 0.020 | +0.4% | 0.496 ± 0.013 | +0.4% |
| BO-qEI | Base | 0.569 ± 0.000 | | 0.883 ± 0.000 | | 0.439 ± 0.000 | | 0.468 ± 0.000 | |
| | IGNITE-2 | 0.569 ± 0.000 | +0.0% | 0.883 ± 0.000 | +0.0% | 0.439 ± 0.000 | +0.0% | 0.467 ± 0.000 | -0.1% |
| | IGNITE | 0.569 ± 0.000 | +0.0% | 0.883 ± 0.000 | +0.0% | 0.439 ± 0.000 | +0.0% | 0.467 ± 0.000 | +0.0% |
| ICT | Base | 0.556 ± 0.024 | | 0.872 ± 0.007 | | 0.564 ± 0.046 | | 0.501 ± 0.018 | |
| | IGNITE-2 | 0.559 ± 0.025 | +0.3% | 0.873 ± 0.007 | +0.1% | 0.554 ± 0.033 | -1.0% | 0.498 ± 0.022 | -0.3% |
| | IGNITE | 0.551 ± 0.022 | -0.5% | 0.878 ± 0.003 | +0.6% | 0.527 ± 0.030 | -3.7% | 0.454 ± 0.038 | -4.7% |

In addition, **IGNITE** also maintains a high probability of $72.73\%$ (32 out of 44) of not degrading baseline performance. There is a $47.73\%$ likelihood (21 out of 44 cases) of improving baseline performance, with an average improvement of approximately $1.0\%$ and a peak improvement of $2.6\%$. Besides, **IGNITE** also exhibits a low probability ($27.27\% = 12$ out of 44 cases) of decreasing performance, with an average degradation of approximately $1.58\%$ and a peak degradation of $4.6\%$. Additionally, there is a small probability ($25\% = 11$ out of 44 cases) of maintaining baseline performance.

### G.3.3 Performance Evaluation at 50-th Percentile Level of IGNITE and IGNITE-2

Table 7 displays the results of **IGNITE-2** and **IGNITE** at the 50th percentile level. The results indicate that both **IGNITE-2** and **IGNITE** do not degrade performance compared to the baseline in $65.91\%$ of settings (29 out of 44) for each method. In which, **IGNITE-2** outperforms the baseline in $50\%$ likelihood, with an average improvement of approximately $0.85\%$ and a peak improvement of $2.9\%$. For **IGNITE** , those are $52.27\%$ likelihood, $1.89\%$ average improvement and a peak improvement of $14.6\%$. Conversely, there are only $34.09\%$ cases in using our proposed method leads to the performance reduction. The average degradation of **IGNITE-2** and **IGNITE** are $0.69\%$ and $1.19\%$, respectively. Overall, we state that our proposed method helps to boost the performance of almost all algorithms and tasks.

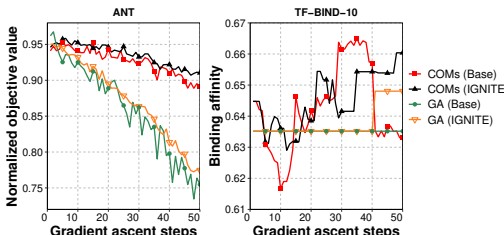

Figure 3: Performance vs. the no. of gradient ascent steps during optimization of **IGNITE-2** and Baseline optimized algorithms, e.g, **COMs** and **GA**.

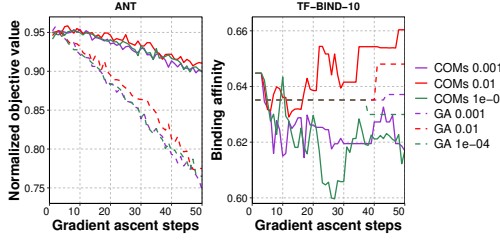

Figure 4: Performance variation of COMS and GA (regularized by **IGNITE-2**) in the change of the regularization coefficient $\lambda \in [0.0001, 0.001, 0.01]$.

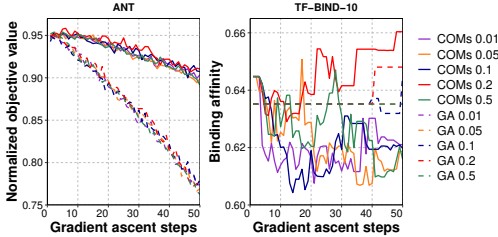

Figure 5: Performance variation of COMS and GA (regularized by **IGNITE-2**) in the change of the hyper-parameter $r \in [0.01, 0.05, 0.1, 0.2, 0.5]$.

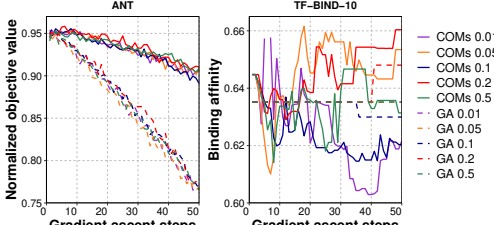

Figure 6: Performance variation of COMS and GA (regularized by **IGNITE-2**) in the change of the hyper-parameter $\rho \in [0.01, 0.05, 0.1, 0.2, 0.5]$.

## G.4 Hyper-parameters selection for IGNITE-2

This section provides the specific setup of the hyper-parameter selection experiments for **IGNITE-2**, which is also ran separately to determine the hyperparameters for **IGNITE**(in the main text).

### G.4.1 IGNITE-2 enhances stability of COMs and gradient ascent (GA).

Figure 3 provides a step-by-step performance comparison between two baseline algorithms, **COMs** and **GA**, with and without our regularization method, **IGNITE-2**. **IGNITE-2** consistently outperforms the baseline algorithms at every step in the **ANT** tasks. However, the trend differs for **TF-BIND-10**. Initially, the baseline algorithms perform better without **IGNITE-2** . However, as the number of gradient ascent steps increases, the performance of the baselines without **IGNITE-2** begins to decline. In contrast, the algorithms incorporating our method maintain or improve their performance over time. This trend suggests that our method becomes increasingly beneficial in the later stages of the optimization process, ultimately enhancing the overall performance of the baseline algorithms.

### G.4.2 Choosing the regularization coefficient $\lambda$.

Figure 4 shows how the performance of baseline models regularized with **IGNITE-2** is affected by different values of $\lambda$. The results indicate that using an excessively low or high value for $\lambda$ will have a negative impact on performance. In all cases, the results suggest that a universal value of 0.01 for $\lambda$ tends to generate consistent and effective performance across all tasks.

### G.4.3 Choosing value for parameter $r$ and the perturbation radius $\rho$.

Figure 5 plot the performance of the **GA** and **COMS** baselines regularized with **IGNITE-2** with respect to the change of hyper-parameter $r$. That is $r \in \{0.01, 0.05, 0.1, 0.2, 0.5\}$. Figure 6 visualizes how the performance of baselines regularized with **IGNITE-2** is influenced by varying the hyperparameter $\rho$. Based on those result, we choose to set $r = 0.2$ and $\rho = 0.2$ in all of the experiments for **IGNITE-2**.

## H  Percentage improvement over other baselines of IGNITE, SAM across all tasks.

Table 8: Percentage improvement over other baselines of **IGNITE**, SAM across all tasks.

| Algorithms | Ant | D'Kitty | TF Bind 8 | TF Bind 10 |
|---|---|---|---|---|
| REINFORCE | 0.255 ± 0.036 | 0.546 ± 0.208 | 0.929 ± 0.043 | 0.635 ± 0.028 |
| REINFORCE + IGNITE | 0.282 ± 0.021 (+2.7%) | 0.642 ± 0.160 (+9.6%) | 0.944 ± 0.030 (+1.5%) | 0.670 ± 0.060 (+3.5%) |
| REINFORCE + SAM | 0.266 ± 0.030 (+1.1%) | 0.625 ± 0.182 (+7.9%) | 0.940 ± 0.035 (+1.1%) | 0.637 ± 0.037 (+0.2%) |
| GA | 0.303 ± 0.027 | 0.881 ± 0.016 | 0.980 ± 0.016 | 0.651 ± 0.033 |
| GA + IGNITE | 0.320 ± 0.044 (+1.7%) | 0.886 ± 0.017 (+0.5%) | 0.985 ± 0.010 (+0.5%) | 0.653 ± 0.043 (+0.2%) |
| GA + SAM | 0.310 ± 0.044 (+0.7%) | 0.868 ± 0.014 (-1.3%) | 0.982 ± 0.015 (+0.2%) | 0.662 ± 0.041 (+1.1%) |
| CbAS | 0.854 ± 0.042 | 0.895 ± 0.012 | 0.919 ± 0.044 | 0.635 ± 0.041 |
| CbAS + IGNITE | 0.859 ± 0.039 (+0.5%) | 0.900 ± 0.015 (+0.5%) | 0.921 ± 0.042 (+0.2%) | 0.652 ± 0.055 (+1.7%) |
| CbAS + SAM | 0.853 ± 0.033 (-0.1%) | 0.897 ± 0.013 (+0.2%) | 0.905 ± 0.053 (-1.4%) | 0.637 ± 0.023 (+0.2%) |
| BO-qEI | 0.812 ± 0.000 | 0.896 ± 0.000 | 0.787 ± 0.112 | 0.628 ± 0.000 |
| BO-qEI + IGNITE | 0.812 ± 0.000 (+0.0%) | 0.896 ± 0.000 (+0.0%) | 0.843 ± 0.109 (+0.3%) | 0.628 ± 0.000 (+0.0%) |
| BO-qEI + SAM | 0.812 ± 0.000 (+0.0%) | 0.896 ± 0.000 (+0.0%) | 0.763 ± 0.098 (-2.4%) | 0.619 ± 0.022 (-0.9%) |

## I  Complexity Overhead of IGNITE.

To analyze the computational complexity of **IGNITE**, we break down the complexity of each step in Algorithm 1.

1. **Initialization (Line 1):** Initializing $\omega^{(1)} \leftarrow \omega^{(0)}$ and $\lambda^{(1)} \leftarrow \lambda$: $O(1)$ each.

2. **Main Loop (Line 2-12):** The loop runs for $T$ iterations. Thus, the complexity of the main loop will be multiplied by $T$.

3. **Sampling (Line 3):** Sampling a batch $\mathcal{B} = \{(\mathbf{x}_i, z_i)\}_{i=1}^{m} \sim \mathcal{D}$: $O(m)$.

4. **Computing $\hat{z}_i$ (Line 4):** Evaluating the surrogate model $g(\mathbf{x}_i; \omega^{(t)})$ for each $i \in [m]$. Assuming the surrogate model evaluation has a computational complexity of $C_g = O(d)$ per sample where $d$ is the number of surrogate parameters, the total complexity is $O(m \cdot d)$.

5. **Computing $g_1$ and $g_2$ (Line 5-6):** Computing gradients $\nabla_\omega \ell(\hat{z}_i, z_i)$ and $\nabla_\omega \hat{z}_i$ have complexities $O(C_\ell)$ and $O(C_{\hat{z}})$ respectively per sample where $C_\ell = C_{\hat{z}} = O(d)$. Therefore, the total complexities are $O(m \cdot d)$.

6. **Computing $\hat{\omega}$ (Line 7):** This involves simple vector operations with complexity $O(d)$, where $d$ is the dimensionality of $\omega$.

7. **Computing $g_3$ (Line 8):** Similar to lines 4 and 6, involving evaluating the surrogate and gradient computations, with complexity $O(m \cdot C_g + m \cdot C_{\hat{z}}) = O(m \cdot d)$.

8. **Computing $g^{(t)}$ (Line 9):** Vector operations involving addition and scalar multiplication with complexity $O(d)$.

9. **Updating $\omega$ (Line 10):** Updating $\omega$ involves simple subtraction operations with complexity $O(d)$.

10. **Updating $\lambda$ (Line 11):** Updating $\lambda$ is an $O(1)$ operation

**Overall Complexity:** Considering the above steps, the most computationally expensive parts are the gradient computations in lines 5, 6, and 8. Thus, the overall complexity per iteration is:

$$O(2m \cdot C_g + m \cdot C_\ell + 2m \cdot C_{\hat{z}})$$

Since this loop runs for $T$ iterations, the total complexity is:

$$O(T \cdot (2m \cdot C_g + m \cdot C_\ell + 2m \cdot C_{\hat{z}}))$$

Furthermore, we have the total complexity of the original baseline is:

$$O(T \cdot (m \cdot C_g + m \cdot C_\ell))$$

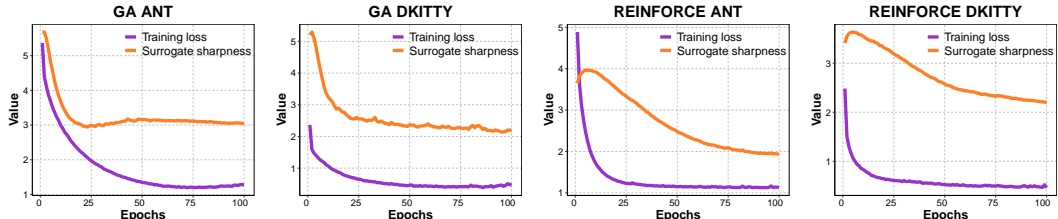

Figure 7: **The convergence of the proposed optimization algorithm.** Figure shows the training loss and the sharpness value (plotting $\|\nabla_\omega h(\omega)\|$ value) during the surrogate fitting process.

Table 9: The empirical time (seconds) of the participating baselines with and without **IGNITE**.

| Algorithms | Ant | D'Kitty | TF Bind 8 | TF Bind 10 |
|---|---|---|---|---|
| REINFORCE | 172.08 | 252.33 | 477.09 | 32.95 |
| REINFORCE + IGNITE | 194.02 (+12.75%) | 275.15 (+9.04%) | 582.28 (+22.05%) | 437.38 (+17.28%) |
| GA | 69.99 | 168.81 | 149.63 | 364.16 |
| GA + IGNITE | 85.15 (+21.66%) | 191.83s (+13.64%) | 181.71 (+21.44%) | 369.29 (+1.41%) |

Thereby, IGNITE will include an additional complexity:

$$O(T \cdot (m \cdot C_g + 2m \cdot C_{\hat{z}})) = O(Tmd)$$

where:

- $T$ is the number of iterations.
- $m$ is the batch size.
- $C_g = O(d)$ is the complexity of evaluating the surrogate model.
- $C_\ell = O(d)$ is the complexity of computing the loss gradient.
- $C_{\hat{z}} = O(d)$ is the complexity of computing the gradient of the surrogate output with respect to its parameters.
- $d$ is the no. of surrogate parameters.

The empirical training time of the participating baselines with and without **IGNITE** is reported in Table 9. We observe that IGNITE solely consumes an additional negligible GPU memory, while the training time increases by 14.91% on average.

## J   Convergence and effectiveness of IGNITE

According to our experiment, despite using a relatively simple BDMM to solve Eq. (12), we have achieved significant improvement in most cases. To demonstrate the convergence of the optimization algorithm, we have plotted the training loss and the sharpness value (plotting $\|\nabla_\omega h(\omega)\|$ value) during the surrogate fitting process. This is based on an experiment using **GA** and **REINFORCE** baselines on **Ant** and **Dkitty** tasks. The results are illustrated in Figure 7. These results reveal that BDMM helps decrease both the training loss and sharpness value of the surrogate model during the training phase. This indicates that BDMM is effective and the optimization converges well in practice. Furthermore, our method, **IGNITE**, can be seamlessly integrated with other, more robust optimization techniques to solve Eq. (12).

## K   Limitation

Our paper studies the offline optimization task, which has potential applications in material engineering. Similar to the existing literature, our method is extensively tested on a universal benchmark set forward by the pioneering work of [4]. However, it is important to note that the benchmark consists mostly of small- to mid-scale optimization tasks. As such, our method has not considered

the challenge of scalability in large-scale domain with extremely high-dimensional input spaces. In addition, as with all machine learning algorithms, while applications of our work to real data could result in ethical considerations, this is an indirect and unpredictable side-effect of our work. Our experimental work uses publicly available datasets to evaluate the performance of our algorithms. No ethical considerations are raised.

