# OpenReview forum: "Incorporating Surrogate Gradient Norm to Improve Offline Optimization Techniques"
_NeurIPS.cc/2024/Conference — NeurIPS 2024 poster_

### Official Review · Reviewer_kVLN · 2024-07-03

**Soundness:** 3
**Presentation:** 3
**Contribution:** 3
**Rating:** 5
**Confidence:** 3

**Summary:**

The article presents a method to improve offline optimization techniques by integrating the concept of model sharpness into the training.  A constraint is introduced that limits the model sharpness to not exceed a user-specified threshold.

**Strengths:**

1. The approach is model-agnostic, making it applicable across different types of models and optimization tasks without needing specific adjustments for each model or task.
2. The method is backed by a solid theoretical analysis, providing a robust framework for understanding and predicting the behavior of the regularization effect on offline optimization.
3. The method has been empirically tested on a variety of tasks, showing significant performance improvements, thus validating the theoretical predictions.

**Weaknesses:**

1. The effectiveness of the method heavily depends on the correct setting of hyperparameters like the sharpness threshold, which can be tricky to optimize without extensive experimentation.
2. The computation complexity is increased. Can the author give the complexity analysis?
3. How to ensure the proposed method converge to a stationary point of the original problem?

**Questions:**

see weakness

---

> ### Author Rebuttal · Authors · 2024-08-07
>
> We would like to thank the reviewer for recognizing the strengths of our work with an acceptance rating.
>
> **Q1. Hyper-parameter Tuning.**
>
> We agree with the reviewer that hyperparameter tuning is important to achieve best performance. This is also true in the broader context of machine learning (ML). Most ML methods require some forms of hyperparameter tuning to achieve good performance. In our case, we adopt the hyperparameter tuning method from Design-Bench [1]. Specifically, we use the previously reported best hyperparameters for each baseline and only tune the additional hyperparameters introduced by the IGNITE regularizer. With a small set of hyperparameters, the tuning cost is not prohibitive.
>
> [1] Gao, Chen, et al. "Design-Bench: Benchmarks for Data-Driven Offline Model-Based Optimization." arXiv preprint arXiv:2202.08450 (2022).
>
> **Q2. Complexity Overhead of IGNITE.**
>
> To analyze the computational complexity of IGNITE, we break down the complexity of each step in Algorithm 1.
>
> 1. **Initialization (Line 1):**
>
>     * Initializing $\omega^{(1)} \leftarrow \omega^{(0)}$ and $\lambda^{(1)} \leftarrow \lambda: O(1)$ each.
>
> 2. **Main Loop (Line 2-12):**
>
>      * The loop runs for $T$ iterations. Thus, the complexity of the main loop will be multiplied by $T$.
>
> 3. **Sampling (Line 3):**
>
>      * Sampling a batch $\mathcal{B} = \{(\mathbf{x}\_i, z\_i)\}\_{i=1}^m \sim \mathcal{D}$: $O(m)$.
>
> 4. **Computing $\hat{z}_i$ (Line 4):**
>
>      * Evaluating the surrogate model $g(\mathbf{x}\_i; \omega^{(t)})$ for each $i \in [m]$. Assuming the surrogate model evaluation has a computational complexity of $C\_g = O(d)$ per sample where $d$ is the number of surrogate parameters, the total complexity is $O(m \cdot d)$.
>
> 5. **Computing $g_1$ and $g_2$ (Line 5-6):**
>
>      * Computing gradients $\nabla\_\omega \ell (\hat{z}\_i, z\_i)$ and $\nabla\_\omega \hat{z}\_i$ have complexities $O(C\_\ell)$ and $O(C\_{\hat{z}})$ respectively per sample where $C\_\ell = C\_{\hat{z}} = O(d)$. Therefore, the total complexities are $O(m \cdot d)$.
>
> 6. **Computing $\hat{\omega}$ (Line 7):**
>
>      * This involves simple vector operations with complexity $O(d)$, where $d$ is the dimensionality of $\omega$.
>
> 7. **Computing $g_3$ (Line 8):**
>
>      * Similar to lines 4 and 6, involving evaluating the surrogate and gradient computations, with complexity $O(m \cdot C\_g + m \cdot C\_{\hat{z}}) = O(m \cdot d)$.
>
> 8. **Computing $g^{(t)}$ (Line 9):**
>      * Vector operations involving addition and scalar multiplication with complexity $O(d)$.
>
> 9. **Updating $\omega$ (Line 10):**
>
>       * Updating $\omega$ involves simple subtraction operations with complexity $O(d)$.
>
> 10. **Updating $\lambda$ (Line 11):**
>
>       * Updating $\lambda$ is an $O(1)$ operation
>
> **Overall Complexity**
>
> Considering the above steps, the most computationally expensive parts are the gradient computations in lines 5, 6, and 8. Thus, the overall complexity per iteration is:
>
> $ O(2m \cdot C\_g + m \cdot C\_\ell + 2m \cdot C\_{\hat{z}})$
>
> Since this loop runs for $T$ iterations, the total complexity is:
>
> $ O(T \cdot (2m \cdot C\_g + m \cdot C\_\ell + 2m \cdot C\_{\hat{z}})) $
>
> Furthermore, we have the total complexity of the original baseline is:
>
> $O(T \cdot (m \cdot C\_g + m \cdot C\_\ell))$
>
> Thereby, IGNITE will include an additional complexity:
>
> $O(T \cdot (m \cdot C\_g + 2m \cdot C\_{\hat{z}} )) = O(Tmd)$
>
> where:
>
> * $T$ is the number of iterations.
>
> * $m$ is the batch size.
>
> * $C\_g = O(d)$ is the complexity of evaluating the surrogate model.
>
> * $C\_\ell = O(d)$ is the complexity of computing the loss gradient.
>
> * $C\_{\hat{z}} = O(d)$ is the complexity of computing the gradient of the surrogate output with respect to its parameters.
>
> * $d$ is the no. of surrogate parameters.
>
>
> The empirical training time of the participating baselines with and without IGNITE is reported in the table below (see Table 1 in the PDF attached to our summary response). This is based on a NVIDIA RTX 3090 GPU with CUDA 11.8 system.
>
> | Algorithms         | Ant               | D'Kitty           | TF Bind 8         | TF Bind 10        |
> |--------------------|-------------------|-------------------|-------------------|-------------------|
> | REINFORCE          | 172.08s           | 252.33s           | 477.09s           | 372.95            |
> | REINFORCE + IGNITE | 194.02s (+12.75%) | 275.15s (+9.04%)  | 582.28 (+22.05%)  | 437.38s (+17.28%) |
> | GA                 | 69.99s            | 168.81s           | 149.63s           | 364.16s           |
> | GA + IGNITE        | 85.15s (+21.66%)  | 191.83s (+13.64%) | 181.71s (+21.44%) | 369.29s (+1.41%)  |
>
> **Q3. How to ensure convergence to an optima of the oracle?**
>
> We would like to emphasize that finding the optima of the oracle is an ill-posed task in the offline context since there might exist infinitely many functions that fit the offline data perfectly but will have different behaviors on the unseen input regions.
>
> Instead, the ultimate goal of offline optimization is to focus the search on a safe region where the output prediction often does not change substantially across surrogate candidates. This can be achieved by finding surrogates with low sharpness. Their gradient will help shape a safe search region within which the surrogate optimum is close to the oracle’s local optimum (not necessarily a stationary point of the oracle). Our contribution here is to develop a rigorous framework to characterize and (provably) control the surrogate sharpness via a non-trivial adaptation of loss sharpness.

---

> > ### Comment · Reviewer_kVLN · 2024-08-10
> > **Response to author**
> >
> > Thank you for the detailed response from the author. The major concern has already been addressed.  I will keep my score.

---

> > > ### Author Response · Authors · 2024-08-10
> > > **Thank you**
> > >
> > > Dear Reviewer kVLN,
> > >
> > > Thank you very much for the prompt response.
> > >
> > > We are glad that our response has addressed your concern.
> > >
> > > Best regards,
> > >
> > > Authors

---

### Official Review · Reviewer_qc6B · 2024-07-09

**Soundness:** 3
**Presentation:** 3
**Contribution:** 3
**Rating:** 7
**Confidence:** 3

**Summary:**

This paper proposes a novel model-agnostic approach to enhance offline optimization methods by incorporating surrogate gradient norms.  The paper provides a thorough review of existing literature, a clear problem definition, and detailed descriptions of the proposed methods and their implementation. The experiments are well-designed, covering various benchmarks and baselines, and the results are robustly analyzed.

**Strengths:**

- a new regularizer based on surrogate sharpness, characterized by the surrogate's maximum output change under low-energy parameter perturbation. This regularizer is designed to be model-agnostic, making it broadly applicable across different surrogate and search models.
- a practical approximation that reduces the surrogate sharpness measurement to a function of the surrogate’s gradient norm. This allows the optimization task to be transformed into a constrained optimization problem, which can be solved using existing optimization solvers.
- a theoretical analysis demonstrating that reducing surrogate sharpness on an offline dataset provably reduces its generalized sharpness on unseen data.
- extensive experimentation on a diverse range of optimization tasks shows that reducing surrogate sharpness often leads to performance improvements.

**Weaknesses:**

- As the paper mentioned that they draw inspiration from the sharpness-aware minimization (SAM), the paper’s novelty could be better articulated in the context of existing work on surrogate model regularization and sharpness-aware optimization. Specifically, discuss how the proposed surrogate sharpness measure offers advantages over sharpness-aware minimization (SAM).
- Assumption 2 (positive minimum eigenvalue of the parameter Hessian) might be too restrictive. It would be more helpful if the authors can provide more intuitive explanations or empirical evidence to justify these assumptions and discuss the implications if these assumptions are violated and how the method’s performance might be affected.
- The proposed method involves computationally expensive operations, such as gradient norm calculations and constrained optimization. I'm wondering how expensive the proposed method is compared to existing base approaches in terms of the training time.

**Questions:**

See above.

**Limitations:**

Yes. In Appendix H

---

> ### Author Rebuttal · Authors · 2024-08-07
>
> We would like to thank the reviewer for recognizing the strengths of our work with an acceptance rating.
>
> **Q1. How surrogate sharpness offers advantages over loss sharpness (SAM).**
>
> We will highlight below an important intuition on the key difference between a direct application of SAM and its non-trivial adaptation to control the surrogate sharpness (rather than its loss sharpness) in offline optimization.
>
> The intuition is that minimizing the loss sharpness guarantees that on average, a single prediction at a randomly selected input in the OOD regime will have low error. However, such errors can accumulate along a gradient search process which results from multiple predictions on sequentially dependent inputs. Fortunately, our intuition in Figure 1 (as elaborated in lines 130-136) suggests that such error accumulation can be mitigated by controlling the surrogate sharpness. The idea is that with a sufficiently large perturbation radius, the oracle will be within the perturbation neighborhood. As surrogates with small sharpness will, by definition, not have their predictions changed substantially within its perturbation neighborhood (including the oracle), their optima will be close to those of the oracle.
>
> As such, keeping the surrogate sharpness small while fitting it to the offline data will lessen the impact of error accumulation in the search phase. This is also verified via our empirical studies reported in Table 2 which compares the impact of using surrogate and loss sharpness on surrogate conditioning for offline optimization.
>
>
> **Q2. Is Assumption 2 strong?**
>
> We note that Assumption 2 can be satisfied for a class of surrogates established in Theorem 1 (see its proof in Appendix A). Since there exists an implementable surrogate formulation for which Assumption 2 holds, we believe it is not a strong assumption.
>
> **Q3. Complexity Overhead of IGNITE.**
>
> To analyze the computational complexity of IGNITE, we break down the complexity of each step in Algorithm 1.
>
> 1. **Initialization (Line 1):**
>
>     * Initializing $\omega^{(1)} \leftarrow \omega^{(0)}$ and $\lambda^{(1)} \leftarrow \lambda: O(1)$ each.
>
> 2. **Main Loop (Line 2-12):**
>
>      * The loop runs for $T$ iterations. Thus, the complexity of the main loop will be multiplied by $T$.
>
> 3. **Sampling (Line 3):**
>
>      * Sampling a batch $\mathcal{B} = \{(\mathbf{x}\_i, z\_i)\}\_{i=1}^m \sim \mathcal{D}$: $O(m)$.
>
> 4. **Computing $\hat{z}_i$ (Line 4):**
>
>      * Evaluating the surrogate model $g(\mathbf{x}\_i; \omega^{(t)})$ for each $i \in [m]$. Assuming the surrogate model evaluation has a computational complexity of $C\_g = O(d)$ per sample where $d$ is the number of surrogate parameters, the total complexity is $O(m \cdot d)$.
>
> 5. **Computing $g_1$ and $g_2$ (Line 5-6):**
>
>      * Computing gradients $\nabla\_\omega \ell (\hat{z}\_i, z\_i)$ and $\nabla\_\omega \hat{z}\_i$ have complexities $O(C\_\ell)$ and $O(C\_{\hat{z}})$ respectively per sample where $C\_\ell = C\_{\hat{z}} = O(d)$. Therefore, the total complexities are $O(m \cdot d)$.
>
> 6. **Computing $\hat{\omega}$ (Line 7):**
>
>      * This involves simple vector operations with complexity $O(d)$, where $d$ is the dimensionality of $\omega$.
>
> 7. **Computing $g_3$ (Line 8):**
>
>      * Similar to lines 4 and 6, involving evaluating the surrogate and gradient computations, with complexity $O(m \cdot C\_g + m \cdot C\_{\hat{z}}) = O(m \cdot d)$.
>
> 8. **Computing $g^{(t)}$ (Line 9):**
>      * Vector operations involving addition and scalar multiplication with complexity $O(d)$.
>
> 9. **Updating $\omega$ (Line 10):**
>
>       * Updating $\omega$ involves simple subtraction operations with complexity $O(d)$.
>
> 10. **Updating $\lambda$ (Line 11):**
>
>       * Updating $\lambda$ is an $O(1)$ operation
>
> **Overall Complexity**
>
> Considering the above steps, the most computationally expensive parts are the gradient computations in lines 5, 6, and 8. Thus, the overall complexity per iteration is:
>
> $ O(2m \cdot C\_g + m \cdot C\_\ell + 2m \cdot C\_{\hat{z}})$
>
> Since this loop runs for $T$ iterations, the total complexity is:
>
> $ O(T \cdot (2m \cdot C\_g + m \cdot C\_\ell + 2m \cdot C\_{\hat{z}})) $
>
> Furthermore, we have the total complexity of the original baseline is:
>
> $O(T \cdot (m \cdot C\_g + m \cdot C\_\ell))$
>
> Thereby, IGNITE will include an additional complexity:
>
> $O(T \cdot (m \cdot C\_g + 2m \cdot C\_{\hat{z}} )) = O(Tmd)$
>
> where:
>
> * $T$ is the number of iterations.
>
> * $m$ is the batch size.
>
> * $C\_g = O(d)$ is the complexity of evaluating the surrogate model.
>
> * $C\_\ell = O(d)$ is the complexity of computing the loss gradient.
>
> * $C\_{\hat{z}} = O(d)$ is the complexity of computing the gradient of the surrogate output with respect to its parameters.
>
> * $d$ is the no. of surrogate parameters.
>
>
> The empirical training time of the participating baselines with and without IGNITE is reported in the table below (see Table 1 in the PDF attached to our summary response). This is based on a NVIDIA RTX 3090 GPU with CUDA 11.8 system.
>
> | Algorithms         | Ant               | D'Kitty           | TF Bind 8         | TF Bind 10        |
> |--------------------|-------------------|-------------------|-------------------|-------------------|
> | REINFORCE          | 172.08s           | 252.33s           | 477.09s           | 372.95            |
> | REINFORCE + IGNITE | 194.02s (+12.75%) | 275.15s (+9.04%)  | 582.28 (+22.05%)  | 437.38s (+17.28%) |
> | GA                 | 69.99s            | 168.81s           | 149.63s           | 364.16s           |
> | GA + IGNITE        | 85.15s (+21.66%)  | 191.83s (+13.64%) | 181.71s (+21.44%) | 369.29s (+1.41%)  |

---

> ### Comment · Reviewer_qc6B · 2024-08-10
> **Thank you**
>
> Thank you for the rebuttal. all my concerns have been addressed by the authors. I've increased my score to support this paper to be accepted.

---

> > ### Author Response · Authors · 2024-08-10
> > **Thank you for increasing the rating**
> >
> > Dear Reviewer qc6B,
> >
> > Thank you very much for increasing the overall rating of our work.
> >
> > We really appreciate your support!
> >
> > Best regards,
> >
> > Authors

---

### Official Review · Reviewer_CQYH · 2024-07-10

**Soundness:** 2
**Presentation:** 3
**Contribution:** 2
**Rating:** 5
**Confidence:** 3

**Summary:**

This paper introduce a sharpness-aware optimization to improve out-of-distribution generalization. While inspired by SAM, the major difference is that this work considers the sharpness of predictor outputs rather than loss landscape. With the proposed notion of sharpness, practical algorithm, IGNITE, is developed through an empirical approximation (Eq. 6) and Taylor approximation of the sharpness constraint.

**Strengths:**

- This material is well written and easy to follow.

- Generalization-aware optimizer is very relevant.

- Technical development appears viable and clear.

**Weaknesses:**

- It is not immediately clear, on an intuition level, why this new notion of sharpness is better than loss sharpness in SAM [8].

- Table 2 only presents SAM on two algorithms (REINFORCE and GA) with no error interval reported. A more throughout benchmarking could improve the empirical evaluation.

**Questions:**

- As illustrated in Figure 1, the intuition is that if the oracle is included in the perturbation neighborhood, a smoother predictor tends to have smaller error. However, Table 1 observes that IGNITE applied to some sub-optimal predictors, for example REINFORCE in Ant Morphology, it still achieves good performance. Since the suboptimal prediction performance may imply that $\hat{\omega}$ is not proximal to $\omega^\star$ hence the local approximation may break in practice. I am curious that why IGNITE should still work in this case, I appreciate if the authors could provide some high-level intuitions.

**Limitations:**

See above.


---
post-rebuttal: -> 5

---

> ### Author Rebuttal · Authors · 2024-08-07
>
> We would like to thank the reviewer for recognizing the clarity and viability of our technical development. Your questions are addressed below.
>
> **Q1. Why the new notion of surrogate sharpness is better than loss sharpness?**
>
> To understand this, note that a surrogate that minimizes its loss sharpness might, on average, have a low prediction error for a single random input sampled from the OOD regime. However, offline optimization requires multiple predictions at consecutive inputs that occur during the gradient search process, where errors can accumulate and lead to suboptimal input candidates.
>
> To mitigate such error accumulation, an alternative approach is to select a surrogate (from those that fit the offline data equally well) with low sharpness. We refer to this as surrogate sharpness. According to our intuition illustrated in Figure 1 and described in lines 130-136, the optima of surrogates with low sharpness tends to be closer to the oracle optima.
>
> This means a well-controlled sharpness of the surrogate landscape (rather than its loss landscape) will help the corresponding gradient search accumulate less error. This is formalized in Eq. (5) which applies a low-sharpness constraint $\mathcal{R}\_\mathcal{X}(\omega) \leq \epsilon^{'}$ to the surrogate fitting loss. Solving Eq. (5) therefore requires upper-bounding $\mathcal{R}\_\mathcal{X}(\omega)$ with a tractable function — see Eq. (6) and Theorem 2.
>
> Such insight is also confirmed via an empirical comparison between IGNITE and a direct application of SAM to minimize the loss sharpness of the surrogate. The reported results in Table 2 show a clear improvement of IGNITE over SAM, which confirms that controlling surrogate sharpness is more beneficial than minimizing loss sharpness.
>
>
> **Q2. More thorough comparison with SAM in Table 2.**
>
> We further run SAM with 2 other baselines BO-qEI and CbAS. In addition, we revise Table 2 by reporting the error interval as in the table below (see Table 4 in the PDF attached to our summary response):
>
>
> | Algorithms         | Ant                   | D'Kitty               | TF Bind 8             | TF Bind 10            |
> |--------------------|-----------------------|-----------------------|-----------------------|-----------------------|
> | REINFORCE          | 0.255 ± 0.036         | 0.546 ± 0.208         | 0.929 ± 0.043         | 0.635 ± 0.028         |
> | REINFORCE + IGNITE | 0.282 ± 0.021 (+2.7%) | 0.642 ± 0.160 (+9.6%) | 0.944 ± 0.030 (+1.5%) | 0.670 ± 0.060 (+3.5%) |
> | REINFORCE + SAM    | 0.266 ± 0.030 (+1.1%) | 0.625 ± 0.182 (+7.9%) | 0.940 ± 0.035 (+1.1%) | 0.637 ± 0.037 (+0.2%) |
> | GA                 | 0.303 ± 0.027         | 0.881 ± 0.016         | 0.980 ± 0.016         | 0.651 ± 0.033         |
> | GA + IGNITE        | 0.320 ± 0.044 (+1.7%) | 0.886 ± 0.017 (+0.5%) | 0.985 ± 0.010 (+0.5%) | 0.653 ± 0.043 (+0.2%) |
> | GA + SAM           | 0.310 ± 0.044 (+0.7%) | 0.868 ± 0.014 (-1.3%) | 0.982 ± 0.015 (+0.2%) | 0.662 ± 0.041 (+1.1%) |
> | CbAS               | 0.854 ± 0.042         | 0.895 ± 0.012         | 0.919 ± 0.044         | 0.635 ± 0.041         |
> | CbAS + IGNITE      | 0.859 ± 0.039 (+0.5%) | 0.900 ± 0.015 (+0.5%) | 0.921 ± 0.042 (+0.2%) | 0.652 ± 0.055 (+1.7%) |
> | CbAS + SAM         | 0.853 ± 0.033 (-0.1%) | 0.897 ± 0.013 (+0.2%) | 0.905 ± 0.053 (-1.4%) | 0.637 ± 0.023 (+0.2%) |
> | BO-qEI             | 0.812 ± 0.000         | 0.896 ± 0.000         | 0.787 ± 0.112         | 0.628 ± 0.000         |
> | BO-qEI + IGNITE    | 0.812 ± 0.000 (+0.0%) | 0.896 ± 0.000 (+0.0%) | 0.843 ± 0.109 (+0.3%) | 0.628 ± 0.000 (+0.0%) |
> | BO-qEI + SAM       | 0.812 ± 0.000 (+0.0%) | 0.896 ± 0.000 (+0.0%) | 0.763 ± 0.098 (-2.4%) | 0.619 ± 0.022 (-0.9%) |
>
>
> The reported performance of the baselines in Table 2 shows that IGNITE surpasses SAM in terms of performance improvement over baselines across most tasks. It can also be observed that IGNITE in general has a smaller error interval than SAM.
>
>
> **Q3. Why does IGNITE still work on base optimizers with poor performance?**
>
>
> We note that each offline optimizer often comprises a predictor component (surrogate) and a search component that navigates the input space using the surrogate gradient to recommend a design candidate. The poor performance of an optimizer therefore does not mean its surrogate is not sufficiently proximal to the oracle.
> Instead, it is possible that the surrogate fits well on the offline data and is reasonably proximal to the oracle (though not perfect) but the search does not recognize well areas where the surrogate prediction is not reliable and gets misled into exploring more in those regions, resulting in sub-optimal performance. In such cases, our insight holds and IGNITE can still improve the surrogate by minimizing its surrogate sharpness. The improved surrogate is more reliable and will lessen the impact of the ineffective search as observed throughout Table 1.
>
> We hope the reviewer would consider increasing the rating if our response above has sufficiently addressed all questions. We will be happy to discuss further if the reviewer has any follow-up questions. Thank you for the detailed feedback.

---

> > ### Author Response · Authors · 2024-08-12
> > **Follow-up**
> >
> > Dear Reviewer CQYH,
> >
> > May we know if our response has addressed your questions?
> >
> > Thank you very much for the interesting questions and suggestions.
> >
> > Best regards,
> >
> > Authors

---

> > > ### Author Response · Authors · 2024-08-13
> > > **Follow-up**
> > >
> > > Dear Reviewer CQYH,
> > >
> > > We hope this message finds you well.
> > >
> > > As the rebuttal discussion will end soon, we really hope to follow up with you on whether our response has addressed your questions sufficiently. Your timely feedback is very valuable for us.
> > >
> > > Thank you very much for your feedback and consideration.
> > >
> > > Best regards,
> > >
> > > Authors

---

> > > > ### Comment · Reviewer_CQYH · 2024-08-13
> > > >
> > > > Thank you for your responses, and the additional experiments! My questions are sufficiently addressed.

---

> > > > > ### Author Response · Authors · 2024-08-13
> > > > > **Thank you very much for increasing the score to 5**
> > > > >
> > > > > Dear Reviewer CQYH,
> > > > >
> > > > > We sincerely appreciate your prompt decision and the supportive comments regarding our work.
> > > > >
> > > > > We are deeply grateful for the time and effort you invested in reviewing our research, and for the valuable feedback that has significantly enhanced our work.
> > > > >
> > > > > Best regards,
> > > > >
> > > > > Authors

---

### Official Review · Reviewer_qZmG · 2024-07-10

**Soundness:** 4
**Presentation:** 3
**Contribution:** 3
**Rating:** 7
**Confidence:** 4

**Summary:**

The paper studies the problem of offline optimization for material design problems. The paper proposes a model-agnostic method that changes the parameters of a model by constraining the sharpness of the model's predictions. Since the model generates smoother predictions, the error between the predictions and the oracle tends to be smaller and therefore the performance of the offline model is potentially more accurate. The paper gives a theoretical analysis and shows that the empirical sharpness can bound the theoretical sharpness. The paper applies the method to various offline optimization method and shows improvements in most of the cases.

**Strengths:**

1. The paper starts with strong intuition, proposes a general optimization objective, and proposes a feasible algorithm to solve the objective efficiently. The paper also gives a theoretical analysis that connects the empirical objective to the theoretical sharpness.

2. The proposed method is model agnostic and therefore the method can be applied to many existing offline optimization methods to get improved performance.

3. The proposed method is shown to achieve good performance across a variety of tasks and is applied with various offline optimization methods.

**Weaknesses:**

1. After certain approximations, the objective is optimized using BDMM. It would be interesting to analyze the specific structures of the objective and see if more effective optimization techniques can be applied. This is also related to the model-agnostic feature of the proposed method. While having the model-agnostic property is convenient, it may also miss the specific structures of the problems/models so that the performance is not ideal. Also, there is no empirical evidence showing whether the optimization is effective or how well the optimization converges in practice.

2. The comparisons with baseline methods are only compared for a limited number of tasks. To fully demonstrate the effectiveness, I think a thorough comparison as done in Table 1 would be necessary.

3. The paper very briefly mentions the background of material design. I think a more comprehensive introduction could be beneficial and probably the paper should add detailed descriptions of the tasks and datasets.

**Questions:**

Please address the weaknesses part.

**Limitations:**

Limitations are properly addressed. No obvious negative societal impact.

---

> ### Author Rebuttal · Authors · 2024-08-07
>
> We would like to thank the reviewer for recognizing our contribution with an acceptance rating.
>
> **Q1.**
>
> **A. More effective optimization techniques.** We agree with the reviewer that an in-depth investigation of the specific structure of the objective could reveal a more effective optimization technique. We will explore this direction in a follow-up of the current work.
>
> **B. Convergence and effectiveness of optimization.** According to our experiment, despite the use of a relatively simple BDMM to solve Eq. (12), we have achieved significant improvement in most cases. To demonstrate the convergence of the optimization algorithm, we have plotted the training loss and the sharpness value (plotting the $|| \nabla_\omega h(\omega) ||$ value) during the surrogate fitting process. This is based on an experiment using GA and REINFORCE baselines on Ant and Dkitty tasks. The results are illustrated in Figure 1 in the attached PDF of our summary response.
>
> These results reveal that BDMM helps decrease both the training loss and sharpness value of the surrogate model during the training phase. This indicates that BDMM is effective and the optimization converges well in practice. Furthermore, our method, IGNITE, can be seamlessly integrated with other, more robust optimization techniques to solve Eq. (12).
>
> **Q2. Comparison in Table 1.**
>
> As explained in the paper, we excluded tasks known for their high inaccuracy and noise in oracle functions from prior works (ChEMBL, Hopper, and Superconductor), as well as those considered excessively expensive to evaluate (NAS). Nevertheless, for a more thorough comparison, we have conducted an experiment running the GA and REINFORCE baselines — with and without being regularized by IGNITE — in Superconductor and Chembl tasks. These results are shown in the table below (Table 3 in the attached PDF in our summary response)
> | Baseline             | Superconductor        | Chembl                |
> |----------------------|-----------------------|-----------------------|
> | GA w/o IGNITE        | 0.514 ± 0.021         | 0.635 ± 0.005         |
> | GA w/ IGNITE         | 0.517 ± 0.011 (+0.3%) | 0.640 ± 0.009 (+0.5%) |
> | REINFORCE w/o IGNITE | 0.471 ± 0.011         | 0.634 ± 0.001         |
> | REINFORCE w/ IGNITE  | 0.492 ± 0.015 (+2.1%) | 0.636 ± 0.008 (+0.2%) |
>
> The table illustrates that IGNITE also boosts the performance of baselines in noisy tasks. With more time and computation resources, we will conduct and include a thorough comparison of these tasks in the appendix.
>
> **Q3. Background Discussion.**
>
> We thank the reviewer for this suggestion. We will revise to include a more detailed description of the tasks and datasets. It will summarize the details of those datasets and tasks in Design-Bench [1]. We will include this information in the appendix.
>
> [1] Gao, Chen, et al. "Design-Bench: Benchmarks for Data-Driven Offline Model-Based Optimization." arXiv preprint arXiv:2202.08450 (2022).

---

### Official Review · Reviewer_tYoe · 2024-07-16

**Soundness:** 3
**Presentation:** 3
**Contribution:** 3
**Rating:** 7
**Confidence:** 4

**Summary:**

This paper introduces a model-agnostic regularization method that reduces surrogate model sharpness to improve generalization in offline optimization. By incorporating a surrogate sharpness measure into the training loss, they provide theoretical proof and extensive experimental validation showing that this approach enhances performance on unseen data, achieving up to a 9.6% improvement in various optimization tasks. The proposed algorithm, IGNITE, demonstrates the effectiveness of this technique across different surrogate models and tasks.

**Strengths:**

1. This paper introduces a novel concept of surrogate sharpness for offline optimization, providing a new robust optimization approach that differs from previous loss-based sharpness-aware minimization methods.
2. Based on surrogate sharpness, the authors propose a new algorithm, IGNITE, and provide corresponding theoretical analysis, demonstrating its ability to bound the worst-case generalized surrogate sharpness.
3. The proposed method can be combined with other offline optimization techniques, consistently enhancing generalization performance when integrated.

**Weaknesses:**

1. The Introduction section does not clearly emphasize the difference between surrogate sharpness and loss sharpness. Figure 1 could be positioned earlier in the paper. Additionally, the annotations and explanations in Figure 1a are somewhat confusing; the concepts of \(\sigma\) and \(\sigma^*\) are not adequately explained in the figure and caption, which could lead to misunderstandings.
2. The Theoretical Analysis section (Section 4) should include more insightful discussions based on Theorems 1 and 2, ideally tying them to the results in the experimental section. The formatting of Equation 16 could also be improved.
3. In the experimental section, the authors mention that the IGNITE algorithm introduces five hyperparameters, but the ablation study only analyzes the effects of a subset of these hyperparameters. Many experimental results are placed in the supplementary materials, but they are not referenced in the main text.

**Questions:**

Please see the weaknesses part.

**Limitations:**

Please see the weaknesses part.

---

> ### Author Rebuttal · Authors · 2024-08-07
>
> We would like to thank the reviewer for recognizing our contribution with an acceptance rating.
>
> **Q1. Introduction Improvement.**
>
> We would like to thank the reviewer for the suggestion. We will clearly emphasize the difference between surrogate sharpness and loss sharpness in the Introduction section. This will allow us to highlight an important intuition on the key difference between a direct application of SAM and its non-trivial adaptation to control the surrogate sharpness (rather than its loss sharpness) in offline optimization.
>
> The intuition is that minimizing the loss sharpness guarantees that, on average, a single prediction at a randomly selected input in the OOD regime will have low error. However, such errors can accumulate along a gradient search process which results from multiple consecutive predictions. Fortunately, our intuition in Figure 1 (as elaborated in lines 130-136) suggests that such error accumulation can be mitigated by keeping the surrogate sharpness small during training. This is also verified via our empirical studies reported in Table 2. We will move Figure 1 to the Introduction to provide an earlier illustration of this insight.
>
> **Notation Clarification.** We will also provide a more detailed explanation of $\delta$ and $\delta_\ast$ in the caption to avoid any confusion. Specifically, $\delta_\ast = \mathrm{argmax}_{||\delta||_2 \leq \rho} |\mathbb{E}[g(x; \omega + \delta)] - \mathbb{E}[g(x; \omega)]| $
>
> **Q2. More insightful discussion.**
>
> We will include a more insightful discussion in the revision. Concretely, we will emphasize the earlier intuition that instead of using a direct application of SAM to minimize the loss sharpness, we adapt SAM to select a surrogate (from those that fit the offline data equally well) with low sharpness.
>
> According to our intuition illustrated in Figure 1 and described in lines 130-136, the optima of surrogates with low sharpness tends to be closer to the oracle optima. This means a well-controlled sharpness of the surrogate landscape (rather than its loss landscape) will help the corresponding gradient search accumulate less error.
>
> This is formalized in Eq. (5) which applies a low-sharpness constraint $\mathcal{R}\_\mathcal{X} (\omega) \leq \epsilon^{'}$ to the surrogate fitting loss. Solving Eq. (5) therefore requires upper-bounding $\mathcal{R}_\mathcal{X}(\omega)$ with a tractable function — see Eq. (6) and Theorem 2.
>
> As Theorem 2 depends on Assumptions 1 & 2, we put forward Theorem 1 to assert that both assumptions can be satisfied with implementable choices of the surrogate, indicating that those are not strong assumptions.
>
> We will also improve the formatting of Eq. (16) for better readability.
>
> **Q3. Ablation studies.**
>
> Due to space constraints, we only presented ablation studies for a subset of the most important hyperparameters. We observed that the performance is less sensitive to changes in the others that were omitted from the ablation studies. In the revision, we will include a more comprehensive ablation study in the appendix and ensure that all previously unreferenced experimental results are appropriately referred to in the main text.

---

> > ### Comment · Reviewer_tYoe · 2024-08-11
> >
> > Thanks for the authors' response. I have no additional questions concerning this paper.

---

> > > ### Author Response · Authors · 2024-08-11
> > > **Thank you**
> > >
> > > Dear Reviewer tYoe,
> > >
> > > Thank you for the response.
> > >
> > > We are glad our response has addressed your questions.
> > >
> > > Best regards,
> > >
> > > Authors

---

### Official Review · Reviewer_WGWE · 2024-07-19

**Soundness:** 3
**Presentation:** 4
**Contribution:** 2
**Rating:** 6
**Confidence:** 3

**Summary:**

The paper proposed IGNITE, a promising method for solving out-of-distribution issue in offline training by introducing model sharpness into the training loss of the surrogate as a regularizer. The key innovation of IGNITE lies on incorperating sharpness-aware minimization into offline training and developing theoretical analysis to show that reducing surrogate sharpness on the offline dataset provably reduces its generalized sharpness on unseen data. Experiments on benchmark datasets show the promising performance.

**Strengths:**

1. Well motivated.
2. Easy to follow. I like the style of writing.
3. Experimental results seem promissing (9.6% performance boost).
4. Solid theoretical analysis are provided.

**Weaknesses:**

Although the paper has many strengths, there are still some weaknesses:
1. Sharpness-aware minimization (SAM) is widely explored in many fields. I think this submission lacks sufficient introducing of SAM, especially on its development. Take some recent works as an example:

Enhancing sharpness-aware optimization through variance suppression.  (NeurIPS'23)
Normalization layers are all that sharpness-aware minimization needs.  (NeurIPS'23)
Domain-Inspired Sharpness-Aware Minimization Under Domain Shifts.  (ICLR'24)
Locally Estimated Global Perturbations are Better than Local Perturbations for Federated Sharpness-aware Minimization.  (ICML'24)

2. The gain on performance is not always stable, as shown in the Table 4 and 5.
3. Lacking sharpness visualization of objectives on unseen data.

**Questions:**

1. I recommend the authors to extent the contends on the related works of sharpness-aware minimization.
2. Can the authors explain the potential reason on the performance drop shown in Table 4 and 5?
3. Can the authors show the sharpness of objectives on unseen data before and after using your algorithm?

**Limitations:**

I don't think there are any negative societal impacts of this work.

---

> ### Author Rebuttal · Authors · 2024-08-07
>
> We would like to thank the reviewer for recognizing the strengths of our work with an acceptance rating.
>
> **Q1. Improving the introduction of SAM.**
>
> We appreciate the reviewer's insightful recommendation. We will cite the suggested works and discuss their impact on the development of SAM in our revision. We will also emphasize in the discussion that a key difference between such development and our work is that we provide a non-trivial adaptation of the proving technique in SAM to characterize and control the surrogate sharpness (rather than its loss sharpness). The intuition is that minimizing the loss sharpness guarantees that, on average, a single prediction at a randomly selected input in the OOD regime will have low error. However, such errors can accumulate along a gradient search process which results from multiple, consecutive predictions. Fortunately, our intuition in Figure 1 (as elaborated in lines 130-136) suggests that such error accumulation can be mitigated by controlling the surrogate sharpness. This is also verified via our empirical studies reported in Table 2 -- see also our response to Q1 of **Reviewer xjU2**.
>
> **Q2. Performance drop in Tables 4 and 5.**
>
> It is expected that the results for the 80th and 50th percentiles are less impressive than those for the 100th percentile. This is because the baseline methods are inevitably less effective at lower percentiles. As seen in Tables 4 and 5, the solutions for the 80th and 50th percentiles from the base methods often show minimal improvement over the empirical best.
>
> **Q3. Sharpness of objectives on unseen data before and after using IGNITE.**
>
> We conducted an experiment to compare the values of surrogate sharpness — $\rho || \nabla_\omega h(\omega) ||$ in Eq. (10) — with and without IGNITE. These surrogate sharpness values were computed on unseen data, which are design candidates found by the GA and REINFORCE baselines before and after being regularized with IGNITE. These are reported in the table below (Table 2 in the attached PDF in our summary response)
>
> | Baseline             | Ant  | Tf-bind-10 |
> |----------------------|------|------------|
> | GA w/o IGNITE        | 1.88 | 1.07       |
> | GA w/ IGNITE         | 1.69 | 0.63       |
> | REINFORCE w/o IGNITE | 0.18 | 0.24       |
> | REINFORCE w/ IGNITE  | 0.09 | 0.14       |
>
> The reported results thus demonstrate that our method, IGNITE, helps decrease the sharpness of the surrogate model on unseen data (as expected).

---

> > ### Comment · Reviewer_WGWE · 2024-08-13
> > **Thanks for the rebuttal**
> >
> > Thanks for the rebuttal. I don't have further questions.

---

> > > ### Author Response · Authors · 2024-08-13
> > > **Thank you**
> > >
> > > Dear Reviewer WGWE,
> > >
> > > Thank you for the response.
> > >
> > > We are glad our response has addressed your concerns.
> > >
> > > Best regards,
> > >
> > > Authors

---

### Official Review · Reviewer_xjU2 · 2024-07-21

**Soundness:** 2
**Presentation:** 2
**Contribution:** 2
**Rating:** 5
**Confidence:** 4

**Summary:**

This paper focuses on offline optimization, presenting a novel approach to enhance the surrogate landscape's sharpness, thereby improving generalization. The authors introduce a gradient norm as an approximation of surrogate sharpness through a first-order Taylor expansion, resulting in a Lagrangian formulation. They employ two widely adopted methods—BDMM and a fixed multiplier as solvers for the Lagrangian.

Theoretically, the authors provide a generalization bound for the worst-case prediction change. Empirically, they validate their approach using the widely adopted offline optimization dataset, design-bench, demonstrating relatively good performance.

**Strengths:**

1. The idea of using a sharpness-aware method in offline optimization is well-motivated.

2. The proposed method performs well on some datasets with specific baselines.

3. The authors provide a generalization analysis.

**Weaknesses:**

Though I agree with the ideas of using a sharpness-aware method in offline optimization, I think more details should be justified:


1. I am somewhat unsure whether the authors analyze the generalization bound on $\mathcal{R}_{X}(\omega)$ properly.

Does the generalization over $R_{X}(\omega)$ help to improve the generalization of $L_{X}(\omega)$, which is the objective we want to generalize. Why not analyze the generalization bound w,r,t $L_{X}(\omega)$?

2. Is the bound tight? I understand that the original SAM paper [8] also provides a similar bound. But if it is not tight, we do not know whether it can guide the design of our algorithm.

3. Moreover, the authors should add more details to their proofs to help readers better understand. For example, in line 416, the authors should present it as a lemma rather than directly stating the results of [8] as there are some assumptions in their proofs which might not be discussed in your proof. In line 458, the authors say that they use Taylor's remainder term, but it seems that the remainder term is directly eliminated. Could the authors discuss more details?

4.  Another concern is that I am not very sure whether Assumption 2 is a strong assumption. If the assumption holds, then the Hessian is positive definite. Since the proof highly relies on it, I am afraid the generalization bound is not established in the general stationary point  (the Hessian is positive semi-definite rather than positive definite).

5.  Though the proposed method shows relatively higher performance than some corresponding baselines on some datasets, the overall improvements are marginal and sometimes even decrease, as shown in table 1. Moreover, different percentages show vastly different performance. For example, reinforce+IGNITE shows a 6.5\% gain over reinforce at the 100th percentile level, but a -4.6\% at the 80th percentile level. How can this be explained? Does it mean that the proposed method is not robust?

6. Is there any analysis on memory and time? The term includes the gradient of the gradient $\nabla_{\omega}|| \nabla_{\omega} h\left(\omega^t\right) ||$, which I am concerned is both time and memory-consuming.


In all, I think the idea is interesting, and the empirical and theoretical results are promising. But I still think more details should be added before I can vote for acceptance.

**Questions:**

Please answer my question mentioned above.

**Limitations:**

Please see the weaknesses part.

---

> ### Author Rebuttal · Authors · 2024-08-07
>
> We thank the reviewer for the detailed feedback. Your questions are addressed below.
>
> **Q1. Why not bounding $\mathcal{L}_\mathcal{X}(\omega)$ directly?**
>
> Bounding $\mathcal{L}\_\mathcal{X}(\omega)$ helps reduce the averaged prediction error in the OOD region but it does not guarantee such errors will not accumulate along consecutive predictions that guide the gradient-based optimization process. This could lead to substantial gaps between its recommended solutions and the true oracle optima.
>
> To elaborate, $\mathcal{L}\_\mathcal{X}(\omega)$ can be bounded with a direct application of Sharpness-Aware Minimization (SAM) [1]. A surrogate that minimizes this bound might have on average a low (single) prediction error in OOD regions. However, offline optimization requires multiple predictions at consecutive inputs that occur during the gradient update process, where errors accumulate and mislead the gradient search toward suboptimal input candidates.
>
> To mitigate this, our intuition in Fig. 1 (see lines 130-136) suggests that the optima of surrogates with low sharpness tend to be closer to the oracle optima. This means a low sharpness value of the surrogate landscape (rather than its loss landscape) will help the gradient search accumulate less error.
>
> This is formulated in Eq. (5) which applies a low-sharpness constraint $\mathcal{R}\_\mathcal{X}(\omega) \leq \epsilon^{'}$ to the surrogate fitting loss. Solving Eq. (5) therefore requires upper-bounding $\mathcal{R}\_\mathcal{X}(\omega)$ with a tractable function — see Eq. (6). This insight is also supported by the empirical studies in Table 2.
>
> [1] Sharpness-aware minimization for efficiently improving generalization. ArXiv:2010.01412.
>
> **Q2. Does the generalization over $\mathcal{R}\_\mathcal{X}(\omega)$ help to improve the generalization of $\mathcal{L}\_\mathcal{X}(\omega)$?**
>
> Following the response to Q1, bounding $\mathcal{L}\_\mathcal{X}(\omega)$ with SAM does not help mitigate the error accumulation along the gradient search. Hence, our approach does not aim to do this.
> Instead, we generalize over $\mathcal{R}\_\mathcal{X}(\omega)$ to solve Eq. (5) which is essential to prevent such error accumulation.
> Investigating whether generalizing over $\mathcal{R}\_\mathcal{X}(\omega)$ also improves generalization over $\mathcal{L}\_\mathcal{X}(\omega)$ is interesting but orthogonal to the scope of our work.
>
> **Q3. Is the bound tight?**
>
> Yes. When the no. of offline data points is sufficiently large, $\frac{\log n}{n}$ tends to zero and Eq. (6) or Eq. (16) in Theorem 2 reduces to $\frac{\mathcal{R}\_\mathcal{X}}{\mathcal{R}\_\mathcal{D}} \leq constant$. This depends only on properties of the surrogate, such as the boundedness of its gradient and the largest eigenvalue of its Hessian. Both can be made small via a theoretic choice of a surrogate whose gradient has a low Lipschitz constant (i.e., smooth).
>
> **Q3. Adding more details to proofs and Taylor’s remainder term in line 458.**
>
> We will present line 416 as a lemma in the revised paper. As for the question regarding the Taylor’s remainder in line 458: it is the second term in Eq. (65) which involves $\hat{\omega} = \omega + c\delta$, which is not eliminated. This is followed by the 1st-order Taylor expansion with an explicit characterization of the remainder term:
>
> $h(\omega + \delta) - h(\omega) = \nabla\_\omega h(\omega)^T \delta + R\_1(\omega, \delta)$ where the remainder $R\_1 (\omega, \delta) = 1/2 \delta^T \nabla^2\_\omega h(\hat{\omega}) \delta$ with $\hat{\omega} = \omega + c\delta$.
>
> For a quick reference, please refer to Eq. (4) in [1], and the 2nd equation in [2].
>
> [1] https://sites.math.washington.edu/~folland/Math425/taylor2.pdf
>
> [2] https://people.math.sc.edu/josephcf/Teaching/142/Files/Worksheets/Estimation%20of%20the%20Taylor%20Remainder.pdf
>
> **Q4. Is Assumption 2 strong?**
>
> Assumption 2 can be satisfied with a class of surrogates established in Theorem 1 (see its proof in Appendix A). Since there exists an implementable surrogate formulation for which Assumption 2 holds, we believe it is not a strong assumption.
>
> **Q5. Is the proposed method (IGNITE) robust?**
>
> The example that the reviewer pointed out – 6.5% gain over REINFORCE at the 100th percentile level, but a -4.6% at the 80th percentile level – does not pertain to our IGNITE method. Instead, it corresponds to a simpler version of our work (IGNITE-2) which is used to demonstrate the impact on performance if we use a manual choice for the Lagrangian parameter in Eq. (12). Our main method (IGNITE) uses BDMM to also optimize this Lagrangian parameter, leading to better and more stable performance than IGNITE-2. It can be observed that overall, IGNITE shows relatively high improvements over the baseline with very few cases of (very) slight performance decrease.
>
> **Q6. Complexity of IGNITE.** According to Algorithm 1 and Appendix F on the effective computation of $\nabla\_\omega ||\nabla\_\omega h(\omega)||$, we provide a time complexity overhead for IGNITE below.
>
> **Complexity Analysis.** Following our detailed complexity analysis in response to Reviewer qc6B, the complexity overhead of IGNITE is  $O(Tmd)$ where:
> * $T$ is the number of iterations.
> * $m$ is the batch size.
> * $d$ is the no. of surrogate parameters.
>
> **Running Time.** We also report the running time of the participating baselines with and without IGNITE in Table 1 in the PDF attached to our summary response.  This is based on a NVIDIA RTX 3090 GPU with CUDA 11.8 system. It can be observed that with IGNITE, the training time increases by 14.91% on average, which is an acceptable overhead in exchange for the significant performance gain. In terms of memory, we also observe that IGNITE incurs a negligible GPU memory overhead.
>
> We hope the reviewer will consider increasing the rating if our response above has addressed all questions sufficiently. We are happy to discuss further if the reviewer has any follow-up questions for us.

---

> > ### Comment · Reviewer_xjU2 · 2024-08-10
> >
> > Thank you for your detailed response. Most of my concerns have been adequately addressed, so I have decided to increase my score.

---

> > > ### Author Response · Authors · 2024-08-10
> > > **Thank you very much for increasing the score to 5**
> > >
> > > Dear Reviewer,
> > >
> > > We sincerely appreciate your prompt decision and your support for our work.
> > >
> > > We are very grateful for the time you dedicated to discussing our research and the valuable feedback that has greatly improved our work.
> > >
> > > Best regards,
> > >
> > > Authors

---

### Author Rebuttal · Authors · 2024-08-07

We would like to thank the AC for securing seven reviews with high quality.

We thank **Reviewers WGWE, tYoe, qZmG, qc6B, and kVLN** for their accepting scores.

We thank **Reviewers xjU2 and CQYH** for the detailed questions, which help us highlight better the key contribution of our work.

Our responses to the reviewers’ main questions are summarized below.

1. We elaborated more on the advantage of surrogate sharpness over loss sharpness, which highlights the conceptual novelty of our work (**tYoe**,**qc6B**, **xjU2**, **CQYH**).
2. We showed that Assumption 2 is not a strong (**qc6B**, **xjU2**) and provided detailed complexity analysis of IGNITE (**qc6B**, **kVLN**, **xjU2**).
3. We showed the reduced surrogate sharpness on unseen data after conditioning with  IGNITE (**WGWE**).
4. We showed the convergence and effectiveness of the BDMM algorithm by plotting the training loss and the sharpness value during the surrogate fitting process (**qZmG**).
5. We articulated on the tightness of the generalization bound (**xjU2**).
6. We explained why IGNITE can help improve even base optimizers with poor performance (**CQYH**).

**Our additional experiment results (in response to some questions from the reviewers) are detailed in the attached PDF.**

We welcome any follow-up questions from the reviewers regarding our rebuttal. We hope that, based on our detailed responses, the reviewers will consider increasing their scores if their concerns have been sufficiently addressed.

---

### Author Response · Authors · 2024-08-10
**Follow-up**

Dear Reviewers,

Thank you again for the detailed reviews.

Please let us know if you have additional questions for us. We will be more than happy to provide further clarification and discussion.

Reviewers CQYH and xjU2:

May we know if our responses have addressed your concerns sufficiently?

We hope our responses have improved your overall assessment of our work.

Best regards,

Authors

---

### Author Response · Authors · 2024-08-14
**Post-rebuttal summary**

Dear AC and reviewers,

We would like to express our gratitude to the Area Chair for coordinating the review process of our paper. Following the rebuttal discussion, we are very happy that all reviewers have recognized our contribution and provided acceptance scores.

We sincerely thank **Reviewers WGWE, tYoe, qZmG, and kVLN** for maintaining their acceptance scores.

We appreciate **Reviewer qc6B** for raising the score to 7 in support of our paper's acceptance.

We are also thankful to **Reviewers xjU2 and CQYH** for increasing their scores toward acceptance.

Overall, based on the post-rebuttal feedback from **Reviewers WGWE, tYoe, kVLN, qc6B, xjU2, CQYH** and the sustained acceptance rating (7) from **Reviewer qZmG**, we believe our rebuttal has addressed all concerns raised by the reviewers.

--

A summary of our responses to address these concerns is provided below:

1. We provided further elaboration on the advantage of surrogate sharpness over loss sharpness, emphasizing the conceptual novelty of our work (**tYoe**,**qc6B**, **xjU2**, **CQYH**).

2. We demonstrated that Assumption 2 is not overly restrictive (**qc6B**, **xjU2**) and provided detailed complexity analysis of IGNITE (**qc6B**, **kVLN**, **xjU2**).

3. We showed the reduction in surrogate sharpness on unseen data after conditioning with IGNITE (**WGWE**).

4. We illustrated the convergence and effectiveness of the BDMM algorithm by plotting the training loss and sharpness value during the surrogate fitting process (**qZmG**).

5. We provided a detailed explanation of the tightness of the generalization bound (**xjU2**).

6. We clarified how IGNITE can enhance even base optimizers with poor performance (**CQYH**).

Once again, we would like to thank the Area Chair and all reviewers for their time and effort in reviewing our paper.

Best regards,

Authors

---

### Decision · Program_Chairs · 2024-09-25

**Decision:**

Accept (poster)

**Comment:**

This paper proposes a technique called IGNITE to regularize surrogate models used in black-box optimization. IGNITE introduces a sharpness constraint on the surrogate model, that computes the maximum distance between the expected outputs of the surrogate model using the current weights $\omega$ and perturbed weights $\omega + \delta$ for perturbations in a neighborhood $\\| \delta \\|_2 \leq \rho$. This ensures that the worst-case prediction change across the perturbation neighborhood is small. This sharpness objective is similar to the one used in sharpness-aware minimization (Foret et al., 2021), but applied to the surrogate model output instead of the loss function.

The authors perform experiments on four tasks from DesignBench (two continuous and two discrete), applying IGNITE to 11 existing optimization techniques (including CMA-ES, REINFORCE, and others). Overall, they find that IGNITE slightly improves over the baselines in most cases. They also provide theoretical analysis showing that reducing the empirical sharpness on offline data will also reduce its sharpness over the data distribution.

This paper received positive reviews, and the authors addressed most of the common points raised by the reviewers. The reviewers thought the paper was well-written and interesting, with good experimental results and theoretical analysis. They also found the proposed method to be well-motivated and model-agnostic.

Overall, this paper brings a useful new technique for regularizing surrogate models that is relevant to the NeurIPS community.